# Alteration of long- and short-term hematopoietic stem cell ratio causes myeloid-biased hematopoiesis

Katsuyuki Nishi[1,2], Taro Sakamaki[1,2], Akiomi Nagasaka[1,2], Kevin Shuolong Kao[3], Kay Sadaoka[1,2], Masahide Asano[4], Nobuyuki Yamamoto[5], Akifumi Takaori-Kondo[6], Masanori Miyanishi[1,2]*

[1]Hematopoietic Stem Cell Biology and Medical Innovation (HSCBMI), Department of Pediatrics, Kobe University Graduate School of Medicine, Kobe, Japan; [2]RIKEN Center for Biosystems Dynamics Research, Kobe, Japan; [3]Weill Cornell, Rockefeller, Sloan-Kettering, Tri-Institutional MD-PhD Program, New York, United States; [4]Institute of Laboratory Animals, Kyoto University Graduate School of Medicine, Kyoto, Japan; [5]Department of Pediatrics, Kobe University Graduate School of Medicine, Kyoto, Japan; [6]Department of Hematology and Oncology, Kyoto University Graduate School of Medicine, Kyoto, Japan

*For correspondence:
miya75@med.kobe-u.ac.jp

## eLife Assessment

This manuscript provides **valuable** insights into the heterogeneity of hematopoietic stem cells and age-associated myeloid-biased hematopoiesis. While several aspects of the study are intriguing and merit further investigation, the current results remain **incomplete** and additional data are necessary to substantiate the conclusions. Some of the methods and data analyses partially support the claims.

**Abstract** Myeloid-biased hematopoiesis is a well-known age-related alteration. Several possibilities, including myeloid-biased hematopoietic stem cell (HSC) clones, may explain this. However, the precise mechanisms remain controversial. Utilizing the Hoxb5 reporter system to prospectively isolate long-term HSCs (LT-HSCs) and short-term HSCs (ST-HSCs), we found that young and aged LT-HSCs co-transplanted into the same recipients demonstrated nearly equivalent myeloid lineage output, contrary to the theory of myeloid-biased HSC clones. Transcriptomics indicated no significant myeloid gene enrichment in aged LT-HSCs compared to their young counterparts. Instead, transplanting reconstituted young HSCs with the ratio of LT/ST-HSCs seen in aged mice can significantly skew the lineage output to myeloid cells. In addition, while the niche environment in the bone marrow minimally affects myeloid-biased hematopoiesis, aged thymi and spleens substantially hinder lymphoid hematopoiesis, resulting in further myeloid domination. Thus, we demonstrate that myeloid-biased hematopoiesis in aged mice originates due to alteration of the ratio between LT-HSCs and ST-HSCs rather than in heterogeneous HSC clones with various cell fates.

## Introduction

Age-associated changes in individuals are deeply correlated with progressive attenuation of cellular functions in tissue stem cells of organs (*Jones and Rando, 2011*). In the hematopoietic system, hematopoietic stem cells (HSCs), which possess self-renewal capacity and multipotency (*Spangrude et al., 1988*; *Weissman and Shizuru, 2008*; *Majeti et al., 2007*), are responsible for various hematopoietic

alterations with age, such as reduced self-renewal capacity (*Morrison et al., 1996*) and myeloid-biased hematopoiesis (*Muller-Sieburg et al., 2004*; *Challen et al., 2010*) due to their functional decline. For example, myeloid-biased hematopoiesis potentially reduces response to infections (*Webster, 2000*), reduces vaccination efficacy (*Crooke et al., 2019*), and increases myeloid malignancy (*Rossi et al., 2008*) in aged individuals. To comprehend these age-associated physiological changes, myeloid-biased hematopoiesis has been studied in considerable detail.

Based on the experimental observation that transplantation of aged HSCs exhibits more myeloid-biased differentiation in young recipient mice than transplantation of young HSCs, this phenotype has been thought to originate in cell-intrinsic changes in the HSC compartment (*Geiger et al., 2013*; *de Haan and Lazare, 2018*; *Schultz and Sinclair, 2016*). A myeloid-biased phenotype after aged HSC transplantation leads to the suggestion that myeloid-biased HSC clones selectively expand with age (*Cho et al., 2008*; *Pang et al., 2011*). Transcriptome and epigenetic analyses showing that a set of genes related to myeloid differentiation is significantly enriched in aged HSCs compared to young HSCs (*Rossi et al., 2005*; *Grover et al., 2016*) further supports this hypothesis.

On the other hand, some reports support a different point of view in light of experimental evidence showing that aged lymphoid-biased HSCs still demonstrate the same level of lymphopoiesis as their younger counterparts, despite exhibiting a myeloid-biased gene expression pattern (*Montecino-Rodriguez et al., 2019*). This result highlights the limitations of predicting a pattern of HSC differentiation based upon gene expression patterns. Moreover, myeloid progenitors such as granulocyte–macrophage progenitor (GMP) and common myeloid progenitor (CMP) should also increase with age if the selective expansion of myeloid-biased HSCs leads to an increase of myeloid cells in peripheral blood (PB). However, the increment of such progenitors with age was not consistently demonstrated in earlier research (*Rossi et al., 2005*; *Rundberg Nilsson et al., 2016*; *Min et al., 2006*). Additionally, a mathematical model demonstrated that aging had no effect on the daily production of CMP supplied from multipotent progenitors (MPPs) (*Busch et al., 2015*; *Dorshkind et al., 2020*).

As such, while selective expansion of myeloid-biased HSC clones is the most widely accepted hypothesis to explain myeloid-biased hematopoiesis in aging (*Mejia-Ramirez and Florian, 2020*; *SanMiguel et al., 2020*), other mechanisms may exist in light of the inconsistent cellular behavior of progenitor fractions relative to HSCs. To the best of our knowledge, no reports have analyzed kinetics of age-associated changes in HSCs, progenitors, and PB cells simultaneously. Therefore, we started our investigation by examining the correlation between age-related changes in PB and bone marrow (BM) to shed light on the mechanism underlying myeloid-biased hematopoiesis that occurs during aging.

## Results

### The discrepancy of age-associated alternation in PB and BM cells into question the existence of a myeloid-biased clone

Mouse HSC research on aging has used mice aged 18 months or older (*Challen et al., 2010*; *Cho et al., 2008*; *Rossi et al., 2005*; *Grover et al., 2016*; *Montecino-Rodriguez et al., 2019*; *Rundberg Nilsson et al., 2016*). However, given that continuous accumulation of cellular stress with age causes a gradual decline of cellular functions, a comprehensive analysis from young to old mice is necessary to unravel mechanisms by which age-associated, myeloid-biased hematopoiesis progresses. Hence, we analyzed changes in PB (*Figure 1—figure supplement 1A*) at multiple time points from young to old mice. These results showed that the percentage of myeloid cells began to change as early as 6 months in mice and continued to increase at a constant rate until the age of ≥23 months (*Figure 1A*).

It has been reported that myeloid-biased hematopoiesis is caused by a selective increase of myeloid-biased clones in the immunophenotypically defined (surface-antigen defined) HSC fraction (*Figure 1—figure supplement 1B*; *Beerman et al., 2010*). According to this hypothesis, age-associated myeloid hematopoiesis progression in PB would be paralleled by an increase in myeloid-biased HSC clones. Therefore, we examined the frequency of the HSC and progenitor fractions in the BM at multiple time points (*Figure 1—figure supplement 1B, C*). We found that the frequency of immunophenotypically defined HSC in BM rapidly increased up to the age of 12 months. After the age, the rate of increase in their frequency appeared to slow down (*Figure 1B*). On the other hand, in contrast to what we

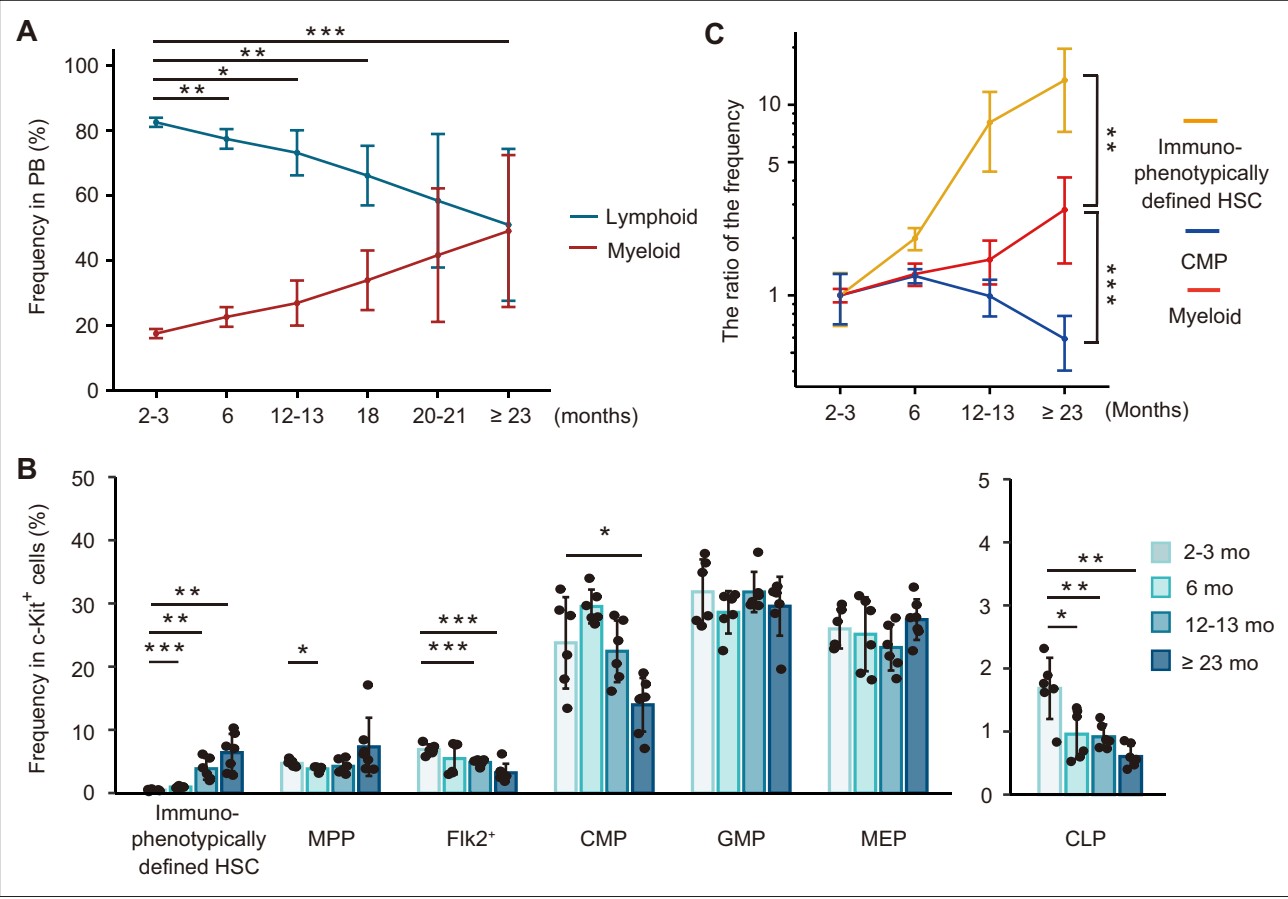

**Figure 1.** Comprehensive analysis of hematopoietic alternations with age shows a discrepancy of age-associated changes between peripheral blood and bone marrow (BM). (**A**) Average frequency of myeloid cells (neutrophils and monocytes) and lymphoid cells (B cells, T cells, and NK cells) in PB at the age of 2 and 3 months ($n = 6$), 6 months ($n = 6$), 12 and 13 months ($n = 6$), 18 months ($n = 6$), 20 and 21 months ($n = 5$), and ≥23 months ($n = 16$). Abbreviation: PB = peripheral blood. (**B**) Average frequency of immunophenotypically defined hematopoietic stem cell (HSC) and progenitor cells in BM of 2- to 3-month mice ($n = 6$), 6-month mice ($n = 6$), 12- to 13-month mice ($n = 6$), and ≥23-month mice ($n = 7$). (**C**) Age-associated changes of immunophenotypically defined HSC and myeloid differentiation components (CMP and myeloid cells in the PB). The ratio of aged to young frequency was calculated as (the fraction frequency at each aged mice (%))/(the average fraction frequency at 2- to 3-month mice (%)). *p < 0.05, **p < 0.01, ***p < 0.001. Data and error bars represent means ± standard deviation.

The online version of this article includes the following figure supplement(s) for figure 1:

**Figure supplement 1.** Gating scheme for hematopoietic cells.

anticipated, the frequency of GMP was stable, and the percentage of CMP actually decreased significantly with age, defying our prediction that the frequency of components of the myeloid differentiation pathway, such as CMP, GMP, and megakaryocyte–erythrocyte progenitor (MEP), may increase in aged mice, if myeloid-biased HSC clones increase with age (*Figure 1B*).

Finally, we determined the ratio of each fraction in young mice versus aged mice to compare the age-associated transition pattern of components comprising the myeloid differentiation pathway. Our analysis of hematopoietic alternations with age revealed that age-associated transition patterns of immunophenotypically defined HSC and CMP in BM were not paralleled with myeloid cell in PB (*Figure 1C*). These findings called into question the hypothesis that there is a selective increase of myeloid-biased HSC clones in the aged BM. We then set out to elucidate the mechanism by which myeloid-biased phenotypes arise in the PB of aged mice.

## The long-term observation of 2-year-old long-term hematopoietic stem cells' differentiation does not indicate the expansion of myeloid-biased clones

Numerous studies have claimed that the myeloid-skewed phenotype observed in PB of aged mice is caused by myeloid-biased HSC clones selectively expanded from HSCs with originally heterogeneous differentiation potentials (*Beerman et al., 2010*; *Dykstra et al., 2011*; *Sudo et al., 2000*). Other studies have reported that immunophenotypically defined HSCs possess heterogeneity associated with self-renewal capacity, suggesting the existence of at least two different populations in the HSC compartment, long-term hematopoietic stem cells (LT-HSCs) and short-term hematopoietic stem cells (ST-HSCs) (*Morrison and Weissman, 1994*; *Spangrude et al., 1995*). Since LT-HSCs have extensive self-renewal capacity, while ST-HSCs lose their self-renewal capacity within a short period, LT-HSCs are thought to persist in the BM throughout life and to enrich age-associated changes compared to ST-HSCs. Therefore, we expect that LT-HSC-specific analyses will help to answer whether myeloid-biased HSC clones exist.

We previously reported that Hoxb5 exclusively marks LT-HSCs in young mice (*Chen et al., 2016*). However, we have not tested in aged mice whether Hoxb5 specifically labels LT-HSCs and helps to distinguish LT-HSCs and ST-HSCs from immunophenotypically defined HSCs (hereafter, bulk-HSCs). To confirm this, we first analyzed expression of Hoxb5 in bulk-HSC, MPP, Flk2$^+$, and Lin$^-$Sca1$^-$c-Kit$^+$ populations in aged *Hoxb5*-tri-mCherry mice. We observed that Hoxb5 was exclusively expressed in bulk-HSCs (*Figure 2A*). Then, to verify long-term engraftment, we conducted a transplantation assay utilizing Hoxb5$^+$ and Hoxb5$^-$ HSCs, respectively, isolated from 2-year-old mice (*Figure 2B*). We observed that only recipients receiving aged Hoxb5$^+$ HSCs exhibited continuous hematopoiesis 16 weeks after primary transplantation (*Figure 2C, E*; *Figure 2—figure supplement 1A*). In secondary transplantation analysis, only recipients receiving Hoxb5$^+$ HSCs exhibited robust hematopoiesis throughout the period of observation, indicating that Hoxb5 can be used as a specific marker of LT-HSCs in aged mice, as well as young mice (*Figure 2D, F*; *Figure 2—figure supplement 1B*).

A serial transplantation assay with a period of observation longer than 8 months should be long enough to observe further myeloid-biased change. If bulk-HSCs isolated from aged mice are already enriched by myeloid-biased HSC clones, we should see more myeloid-biased phenotypes 16 weeks after primary and the secondary transplantation. However, we found that the kinetics of the proportion of myeloid cells in PB were similar across the primary and the secondary transplantation (*Figure 2G*). These results suggest the following two possibilities: either myeloid-biased HSCs do not expand in the LT-HSC fraction, or the expansion of myeloid-biased clones in 2-year-old mice has already peaked.

## Direct comparison between young versus aged LT-HSC differentiation reveals that LT-HSCs exhibit unchanged lineage output throughout life

We developed a co-transplantation assay to directly compare the differentiation capacity between young and aged LT-HSCs by co-transplanting 10 young GFP$^+$ LT-HSCs and 10 aged GFP$^-$ LT-HSCs into the same recipient mice (*Figure 3A*). Then, we found that the myeloid lineage proportions from young and aged LT-HSCs were nearly comparable during the observation period after transplantation (*Figure 3B, C*). Furthermore, by analyzing the proportion of mature cell types derived from young and aged LT-HSCs in the same donor, we directly compared the capacity for hematopoietic reconstitution in each mature cell type between young and aged LT-HSCs. We confirmed again that the reconstitution ratio was almost the same across all lineages, although bulk hematopoiesis derived from young LT-HSCs predominated (*Figure 3D*). These results indicate that the differentiation potential of LT-HSCs remains unchanged throughout their lives.

Several reports have demonstrated that transplanting mixed/bulk-HSCs—a combination of LT-HSCs and ST-HSCs—obtained from old animals results in blatantly myeloid-biased hematopoiesis (*Beerman et al., 2010*; *Dykstra et al., 2011*; *Sudo et al., 2000*). To test this, we co-transplanted bulk-HSCs from young and old mice to corroborate this (*Figure 3—figure supplement 1A*). As previously described, we observed that aged bulk-HSC exhibited a myeloid-skewed phenotype (*Figure 3—figure supplement 1B, C*). Additionally, we observed that aged bulk-HSC reconstitution exhibited higher reconstitution of myeloid cells compared to young bulk-HSCs (*Figure 3—figure supplement 1D*). These findings unmistakably demonstrated that mixed/bulk-HSCs showed myeloid-skewed hematopoiesis in

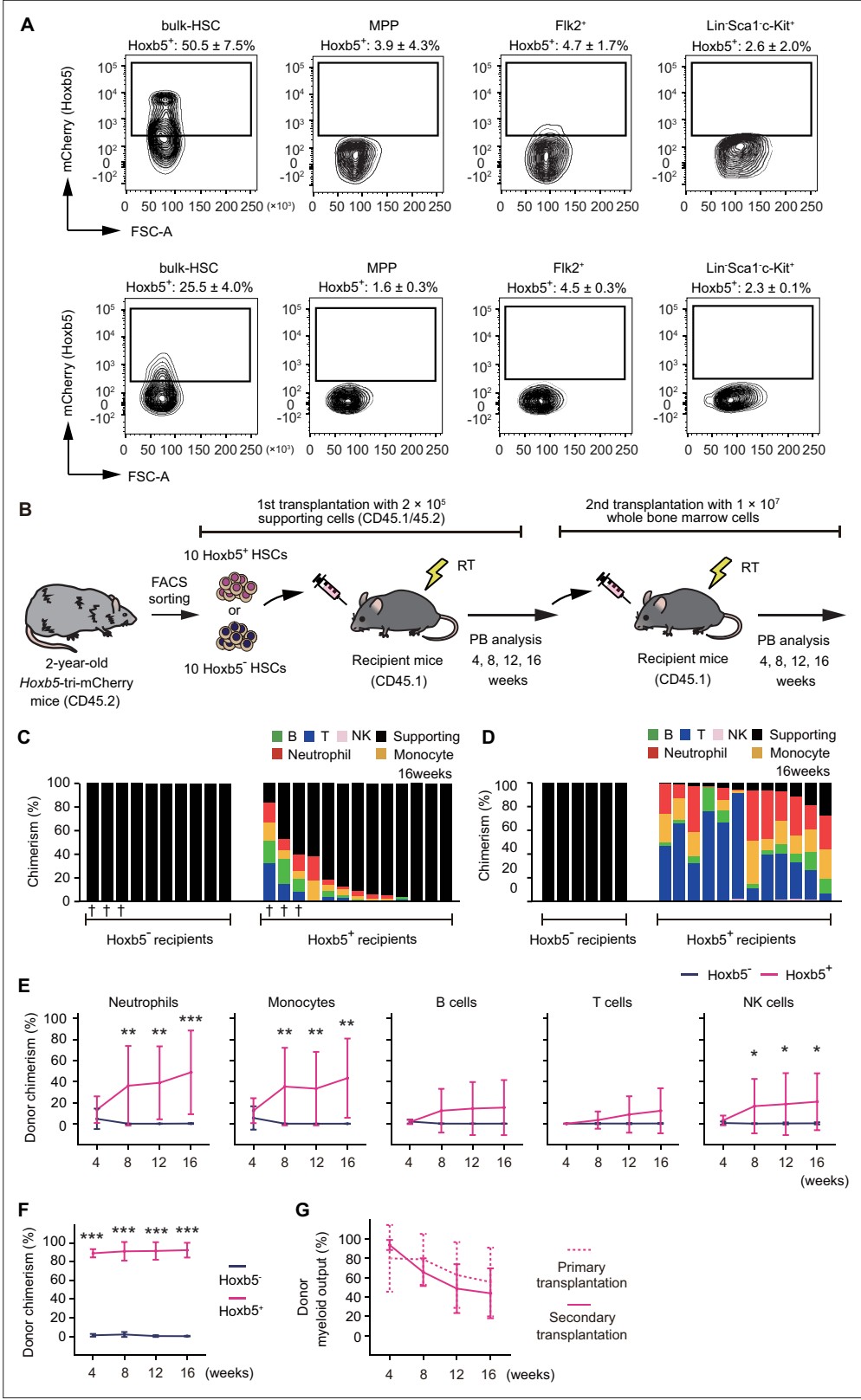

**Figure 2.** The expansion of myeloid-biased clones was not observed in 2-year-old long-term hematopoietic stem cells (LT-HSCs) after their transplantation. (**A**) Hoxb5 reporter expression in bulk-hematopoietic stem cell (HSC), MPP, Flk2+, and Lin−Sca1−c-Kit+ populations in the 2-year-old *Hoxb5*-tri-mCherry mice (Upper panel) and 3-month-old *Hoxb5*-tri-mCherry mice (lower panel). Values indicate the percentage of mCherry+ cells ± standard

*Figure 2 continued on next page*

*Figure 2 continued*

deviation in each fraction (*n* = 3). (**B**) Experimental design to assess the long-term reconstitution ability of Hoxb5$^+$ or Hoxb5$^-$ HSCs. Hoxb5$^+$ and Hoxb5$^-$ HSCs were isolated from 2-year-old CD45.2 *Hoxb5*-tri-mCherry mice and were transplanted into lethally irradiated CD45.1 recipient mice with 2 × 10$^5$ supporting cells (Hoxb5$^+$ HSCs, *n* = 13; Hoxb5$^-$ HSCs, *n* = 10). For secondary transplants, 1 × 10$^7$ whole bone marrow (BM) cells were transferred from primary recipient mice. Abbreviations: PB = peripheral blood, RT = radiation therapy. (**C**) Percentage chimerism at 16 weeks after receiving 10 aged Hoxb5$^-$ HSCs or 10 aged Hoxb5$^+$ HSCs. Each column represents an individual mouse. (**D**) Percentage chimerism at 16 weeks after whole BM secondary transplantation. Donor whole BM cells for secondary transplantation were taken from mice denoted by † in (**C**). (**E**) Kinetics of average donor chimerism in each PB fraction after primary transplantation. (**F**) Kinetics of average donor chimerism after secondary transplantation. (**G**) Kinetics of average donor myeloid output myeloid proportion in donor cells in LT-HSC recipient mice after primary and secondary transplantation. *p < 0.05, **p < 0.01, ***p < 0.001. Data and error bars represent means ± standard deviation.

The online version of this article includes the following figure supplement(s) for figure 2:

**Figure supplement 1.** Aged Hoxb5$^+$ hematopoietic stem cells (HSCs) retain robust hematopoietic capacity.

PB with aging. In contrast, LT-HSCs maintained a consistent lineage output throughout life, although subtle differences between aged and young LT-HSCs may exist and cannot be entirely ruled out.

## LT-HSCs never show myeloid-related gene set enrichment during aging

Although the results of co-transplantation of young and aged LT-HSCs did not demonstrate lineage-biased output by LT-HSCs throughout life, enrichment of myeloid genes in aged bulk-HSCs shown by previous studies supports the idea that myeloid-biased hematopoiesis is caused by selective expansion of myeloid-biased HSC clones (*Rossi et al., 2005*; *Grover et al., 2016*). To compare age-associated myeloid gene enrichment, we isolated bulk-HSC, LT-HSCs, and ST-HSCs from young or aged mice (*Figure 4A*).

To ensure the quality of the sorted fraction for subsequent RNA-seq analyses, we verified the Hoxb5 read counts (*Figure 4—figure supplement 1A*). Cluster dendrograms using the whole transcriptome confirmed that cell fractions isolated from young and aged mice clustered into distinct groups (*Figure 4B*). Then, using sets of aging-related genes, such as inflammation (*Liberzon et al., 2015*; *Pietras, 2017*), DNA damage (p53 pathway) (*Liberzon et al., 2015*; *Rossi et al., 2007a*), cell cycle progression (E2F target) (*Liberzon et al., 2015*; *Kowalczyk et al., 2015*), and a common aging signature (*Flohr Svendsen et al., 2021*), we ran a violin plot analysis on each cell fraction to confirm the occurrence of typical aging-related gene expression changes. We found age-associated changes in inflammation and the DNA damage between cells isolated from young and aged mice, but very similar patterns regardless of cell fraction in young or aged mice. In terms of cell cycle progression and aging signature genes, however, we discovered that only the young LT-HSC fraction differed from other young fractions and tended to represent similar gene expression patterns with aged fractions (*Figure 4C*). Then, taking LT-HSC-specific gene expression pattern into account, we looked at expression patterns of specific genes that had been highly validated by experiments as being associated with myeloid-biased HSC (*Beerman et al., 2010*; *Flohr Svendsen et al., 2021*; *Sanjuan-Pla et al., 2013*; *Mann et al., 2018*). We discovered that the expression of these genes is relatively comparable between young and aged LT-HSCs, but not other fractions (*Figure 4—figure supplement 1B*).

A recent comprehensive analysis of mouse HSC aging using multiple RNA-seq datasets claimed that almost 80% of differentially expressed genes are poorly reproducible across datasets (*Flohr Svendsen et al., 2021*). In fact, we found that almost 80% of genes are not shared when we compare three representative myeloid/lymphoid gene sets (*Sanjuan-Pla et al., 2013*; *Chambers et al., 2007*; *Pronk et al., 2007*). Additionally, most genes were only used in a single gene set (*Figure 4D, E*). By using only genes that were shared in these gene sets, we ran a gene set enrichment analysis (GSEA) to see whether myeloid/lymphoid genes were enriched in aged LT-HSCs or other fractions. Neither aged LT-HSCs nor aged ST-HSCs exhibited myeloid/lymphoid gene set enrichment, while shared myeloid-related genes tended to be enriched in aged bulk-HSCs, although this enrichment was not statistically significant (*Figure 4F, G*). On the other hand, GSEA analysis using original gene sets, respectively, showed inconsistent results (*Figure 4—figure supplement 1C, D*).

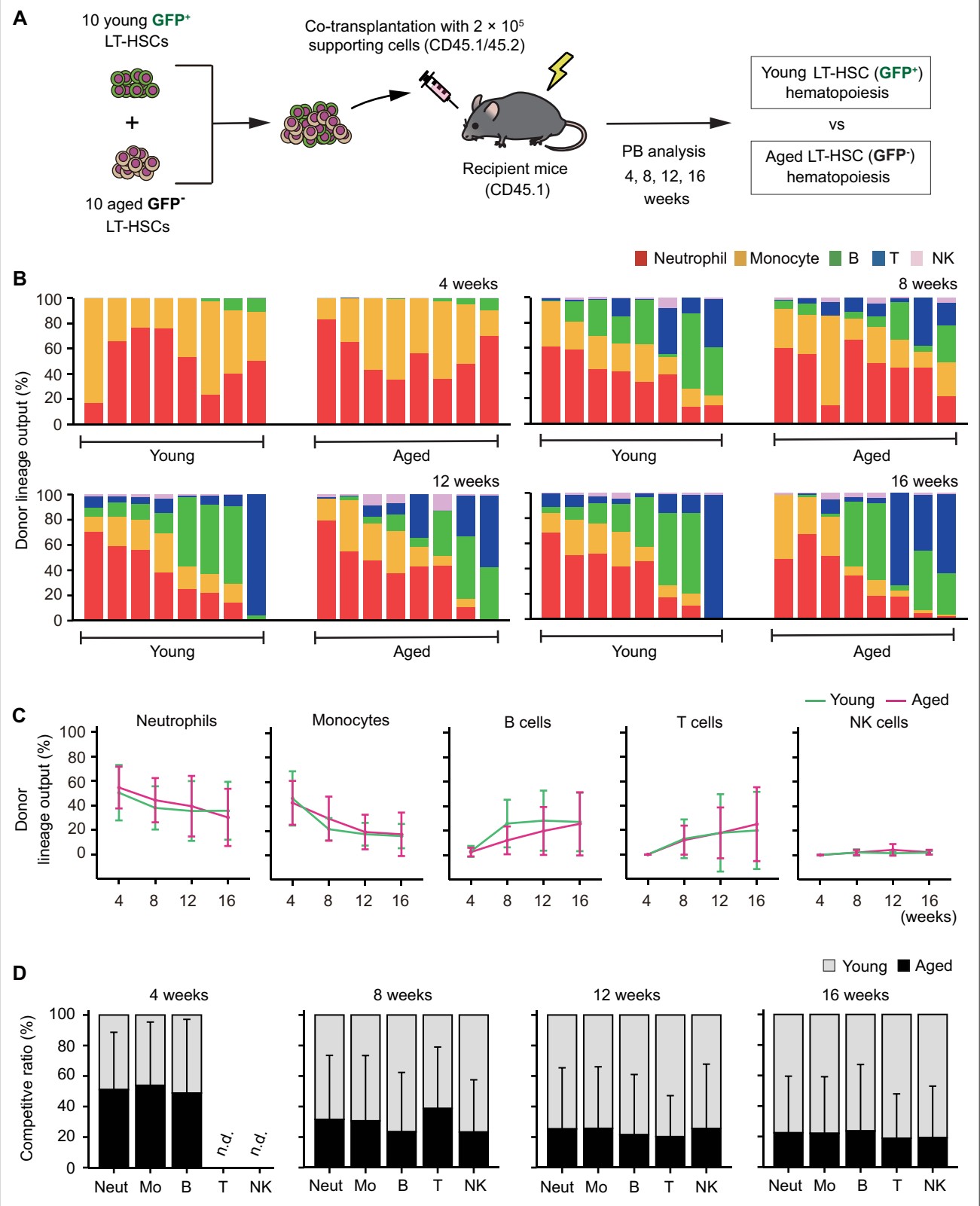

**Figure 3.** Aged long-term hematopoietic stem cells (LT-HSCs) show balanced hematopoiesis throughout life. (**A**) Experimental design for competitive co-transplantation assay using young LT-HSCs sorted from *Hoxb5*-tri-mCherry GFP mice and aged LT-HSCs sorted from *Hoxb5*-tri-mCherry mice. Ten CD45.2[+] young LT-HSCs and 10 CD45.2[+] aged LT-HSCs were transplanted with 2 × 10[5] CD45.1[+]/CD45.2[+] supporting cells into lethally irradiated CD45.1[+] recipient mice (*n* = 8). (**B**) Lineage output of young or aged LT-HSCs at 4, 8, 12, and 16 weeks after transplantation. Each bar represents an individual

*Figure 3 continued on next page*

*Figure 3 continued*

mouse. (**C**) Lineage output kinetics of young LT-HSCs or aged LT-HSCs at 4, 8, 12, and 16 weeks post-transplant. (**D**) Competitive analysis of young LT-HSCs versus aged LT-HSCs lineage output at 4, 8, 12, and 16 weeks post-transplant. The competitive ratio was calculated as the proportion of young LT-HSC-derived cells versus aged LT-HSC-derived cells in each fraction. Abbreviations: Neut = neutrophils, Mo = monocytes, B = B cells, T = T cells, and NK = NK cells. Data and error bars represent means ± standard deviation. 'n.d.' stands for 'not detected'.

The online version of this article includes the following figure supplement(s) for figure 3:

**Figure supplement 1.** Aged bulk-hematopoietic stem cells (HSCs) show myeloid-biased hematopoiesis compared to young bulk-HSCs.

## A myeloid-biased phenotype in PB depends on the relative decrease of ST-HSC in the HSC compartment with age

While transplantation of aged bulk-HSCs exhibits a myeloid-biased phenotype in PB shown in previous reports (*Beerman et al., 2010*; *Dykstra et al., 2011*; *Sudo et al., 2000*) and our results (*Figure 3— figure supplement 1*), aged LT-HSCs remain stable in terms of the balance of their differentiation potential between myeloid and lymphoid production (*Figure 3*). Comparative transcriptome analyses demonstrated that neither LT-HSCs nor ST-HSCs, which are functionally more homogeneous than bulk-HSCs, exhibited myeloid gene set enrichment with age, whereas aged bulk-HSCs tended to show more myeloid gene set enrichment than their young counterparts (*Figure 4*). We then tried to figure out what causes the myeloid-biased phenotype in PB after transplantation of bulk-HSCs.

Because LT-HSCs in bulk-HSCs exhibit nearly constant levels of hematopoiesis throughout life, we hypothesized that ST-HSCs could be the key to discovering mechanisms underlying the myeloid-skewing phenomenon. First, we revisited our previous study (*Chen et al., 2016*), which demonstrated that transplanted ST-HSCs maintain lymphocyte production while rapidly losing myeloid lineage production in recipients. This result may potentially indicate the presence of HSC clones with a preference for lymphoid differentiation. Given that donor cells obtained from primary recipients receiving ST-HSCs do not undergo hematopoiesis in secondary transplantation (*Chen et al., 2016*), it is likely that all ST-HSCs have already lost their ability to self-renew and have disappeared from the HSC fraction in primary recipients. To confirm this, we transplanted 10 LT-HSCs or ST-HSCs and then analyzed PB and BM (*Figure 5A*). As we previously reported (*Chen et al., 2016*), donor-derived hematopoiesis eventually becomes lymphocyte dominant in PB of ST-HSC recipients over time (*Figure 5B*). Subsequently, it was determined that the vast majority of T cell subsets are memory cells (*Figure 5C, D*). In addition, unlike recipients receiving LT-HSCs, recipients receiving ST-HSCs lacked any bulk-HSCs in their BM (*Figure 5E, F*). These results strongly suggest that lymphoid-biased hematopoiesis observed after transplantation of ST-HSCs is due to persistence of memory-type lymphocytes in PB, rather than to de novo lymphopoiesis from lymphoid-biased HSC clones. Persistence of memory-type lymphocytes from ST-HSCs may have led us to misinterpret mechanisms underlying myeloid-biased hematopoiesis after aged bulk-HSC transplantation. To verify this hypothesis, we examined the kinetics of the LT-HSC/ST-HSC ratio in the bulk-HSC population, revealing the ratio of ST-HSC to LT-HSC decreased with age (*Figure 5—figure supplement 1*).

Thus, we hypothesized that the relative decrease in the ST-HSC ratio in the aged bulk-HSC fraction would lead to reduction of memory-type lymphocytes and myeloid-biased hematopoiesis following transplantation of aged bulk-HSCs. To test this hypothesis, we isolated LT-HSCs and ST-HSCs from young donor mice and reconstituted them with a 2:8 ratio (as in young mice) or a 5:5 ratio (as in aged mice) prior to transplanting them (*Figure 6A*). Four weeks after transplantation, PB analysis revealed that both groups had comparable patterns of hematopoiesis. In contrast, recipient mice transplanted with a 5:5 ratio began to exhibit more myeloid-biased hematopoiesis 8 weeks after transplantation, and by week 16, they produced significantly more myeloid cells than the other group (*Figure 6B–D*). Additionally, we conducted further investigations to determine whether lymphoid hematopoiesis could be accelerated by alteration of LT-HSC/ST-HSC ratios using cells from aged mice. To verify this, we isolated aged LT-HSCs and ST-HSCs and reconstituted them with a 5:5 ratio or a 2:8 ratio prior to transplantation (*Figure 6—figure supplement 1A*). PB analysis revealed that lymphoid lineage output in the recipient mice transplanted with a 2:8 ratio was significantly greater than those with a 5:5 ratio (*Figure 6—figure supplement 1B–D*). Based on these findings, we concluded that myeloid-biased hematopoiesis observed following transplantation of aged HSCs was caused by a relative decrease in

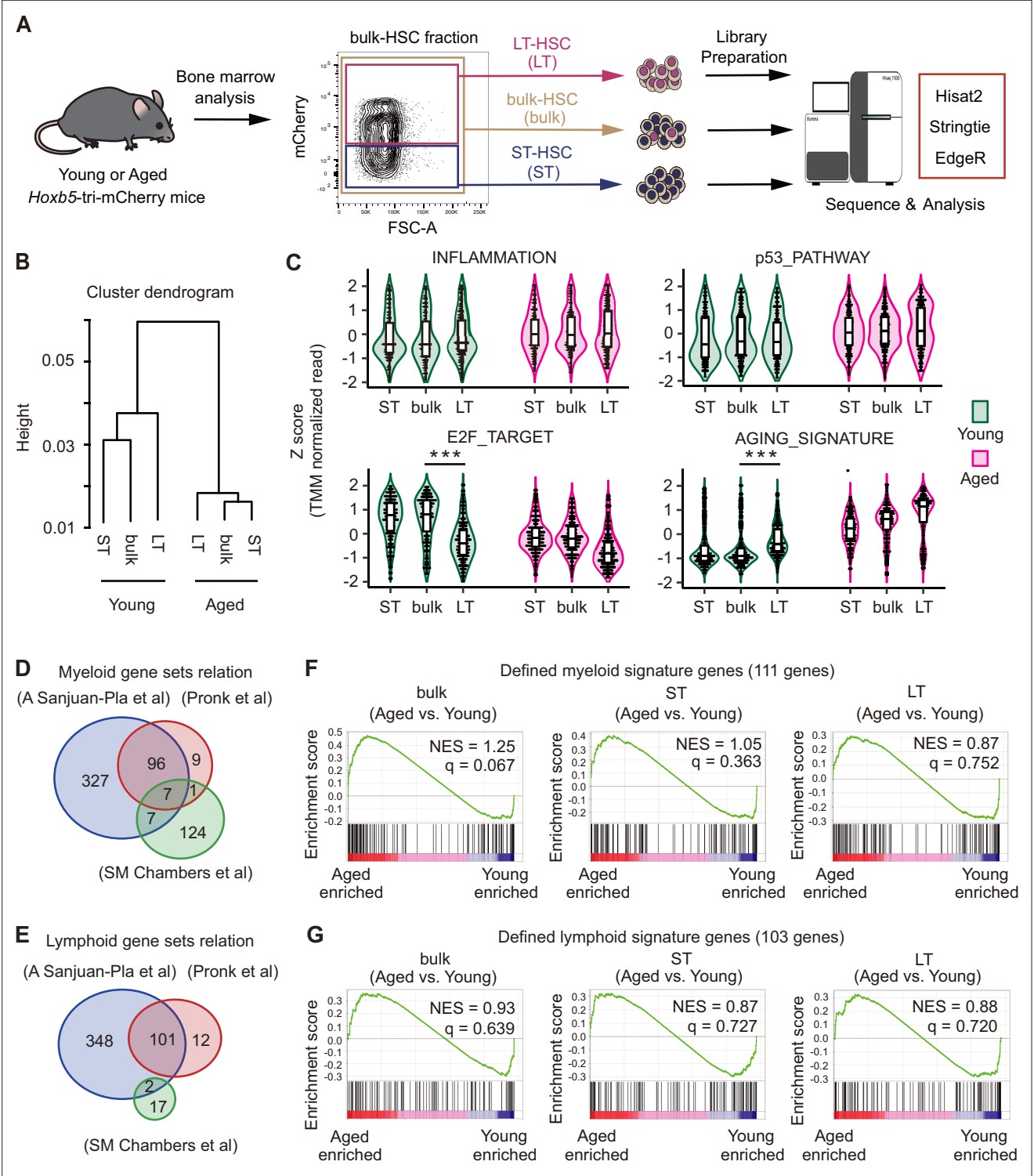

**Figure 4.** Myeloid-associated genes were not enriched in aged long-term hematopoietic stem cells (LT-HSCs) compared to their young counterparts. (**A**) Experimental schematic for transcriptome analysis. LT-HSCs (*n* = 3), short-term hematopoietic stem cells (ST-HSCs) (*n* = 3), and bulk-hematopoietic stem cells (HSCs) (*n* = 3) were sorted from young (2–3 months) or aged (23–25 months) *Hoxb5*-tri-mCherry mice, after which each RNA was harvested for RNA sequencing. (**B**) Hierarchical clustering dendrogram of whole transcriptomes using Spearman distance and the Ward clustering algorithm. (**C**) Violin plots showing normalized gene expression levels for each gene set in young and aged LT-HSCs, ST-HSCs, and bulk-HSCs. Expression values for each gene were standardized independently by applying *Z* score transformation. (**D, E**) Venn diagram showing the overlap of genes between three myeloid signature gene sets and lymphoid signature gene sets (*Sanjuan-Pla et al., 2013*; *Pronk et al., 2007*; *Chambers et al., 2007*). (**F, G**) Signature

*Figure 4 continued on next page*

*Figure 4 continued*

enrichment plots from gene set enrichment analysis (GSEA) using defined myeloid and lymphoid signature gene sets that overlapped in the three gene sets. Values indicated on individual plots are the normalized enrichment score (NES) and *q*-value of enrichment. ***p < 0.001.

The online version of this article includes the following figure supplement(s) for figure 4:

**Figure supplement 1.** Compared to their respective young controls, aged bulk-hematopoietic stem cells (HSCs) exhibit greater enrichment of myeloid gene expression than aged long-term hematopoietic stem cells (LT-HSCs).

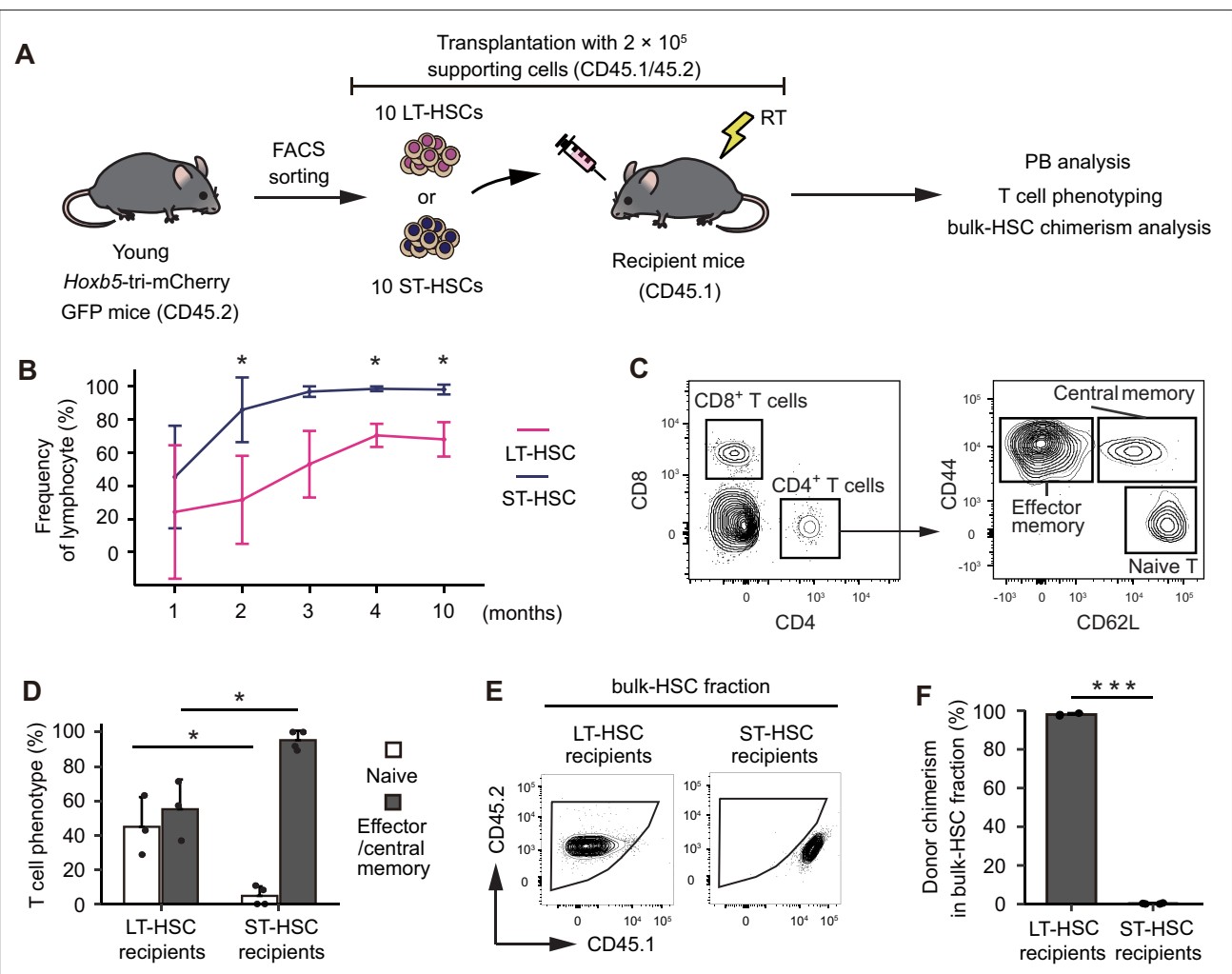

**Figure 5.** The memory-type lymphocytes in the peripheral blood (PB) make it look as if short-term hematopoietic stem cells (ST-HSCs) are lymphoid-biased hematopoietic stem cells (HSCs). (**A**) Experimental design for assessing the lineage output of young long-term hematopoietic stem cells (LT-HSCs) or ST-HSCs. Ten LT-HSCs or 10 ST-HSCs were isolated from 2-month-old CD45.2 *Hoxb5*-tri-mCherry GFP mice and were transplanted into lethally irradiated CD45.1 recipient mice with $2 \times 10^5$ supporting cells (LT-HSCs, n = 3; ST-HSCs, n = 4). (**B**) Kinetics of average frequency of lymphoid cells (B cells, T cells, and NK cells) in donor fraction after LT-HSC or ST-HSC transplantation. (**C**) Gating scheme to identify memory (central and effector) T cells and naive T cells in the PB after excluding doublets, dead cells, and non-donor cells. (**D**) Percentage of memory (central and effector) T cells and naive T cells in donor CD4+ fraction 10 months after LT-HSC or ST-HSC transplantation. (**E**) Gating scheme to identify donor cells in bulk-HSC fraction in bone marrow analysis. (**F**) Donor chimerism in bulk-HSC fraction 12 months after LT-HSC or ST-HSC transplantation. *p < 0.05, ***p < 0.001. Data and error bars represent means ± standard deviation.

The online version of this article includes the following figure supplement(s) for figure 5:

**Figure supplement 1.** Short- and long-term hematopoietic stem cell (ST-HSC and LT-HSC) ratio within bulk-hematopoietic stem cell (HSC) fraction changes with age.

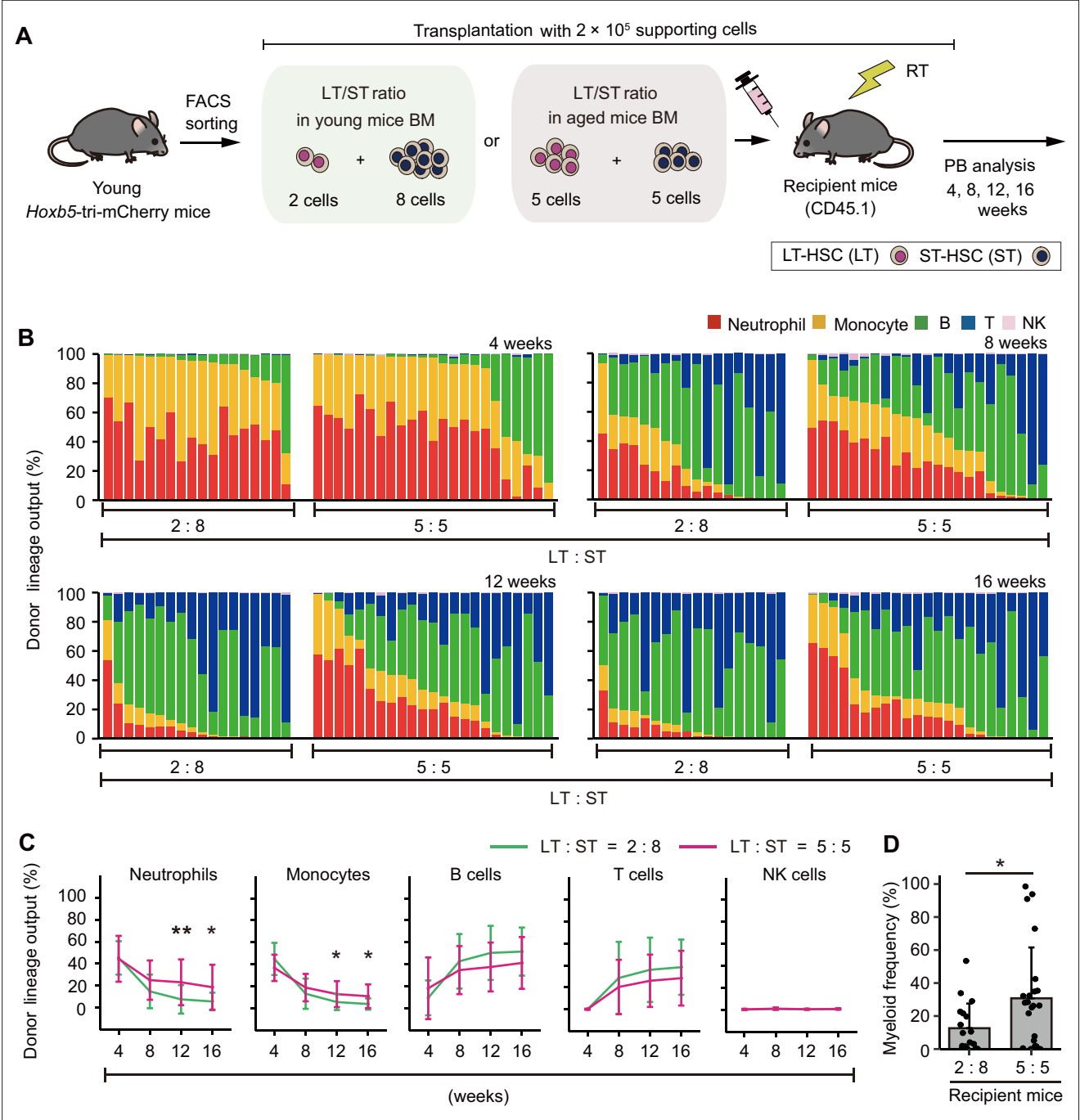

**Figure 6.** Hematopoiesis after transplantation inclined either toward myeloid or lymphoid cell production by artificially changing the ratio of long-term hematopoietic stem cell (LT-HSC)/short-term hematopoietic stem cell (ST-HSC). (**A**) Experimental design for the transplantation of 2- to 3-month-old LT-HSCs and ST-HSCs in a 2:8 ratio (the same ratio as in young mice bone marrow [BM]) or 5:5 ratio (the same ratio as in aged mice BM). Donor cells were transplanted with $2 \times 10^5$ CD45.1$^+$/CD45.2$^+$ supporting cells into lethally irradiated CD45.1$^+$ recipient mice (2:8 ratio, $n = 18$; 5:5 ratio, $n = 23$). (**B**) Donor lineage output of young LT-HSC and ST-HSC transplanted either in a 2:8 ratio or a 5:5 ratio at 4, 8, 12, and 16 weeks post-transplant. Each bar represents an individual mouse. (**C**) Kinetics of average lineage output of young LT-HSCs and ST-HSCs in a 2:8 ratio or a 5:5 ratio at 4, 8, 12, and 16 weeks post-transplant. (**D**) Frequency of myeloid cells in donor cell fraction. *$p < 0.05$, **$p < 0.01$. Error bars represent standard deviation. Data represent two independent experiments.

The online version of this article includes the following figure supplement(s) for figure 6:

**Figure supplement 1.** Changing the short-term hematopoietic stem cell (ST-HSC)/long-term hematopoietic stem cell (LT-HSC) ratio accelerates lymphoid hematopoiesis in the use of aged donor cells.

ST-HSC in the bulk-HSC compartment in aged mice rather than the selective expansion of myeloid-biased HSC clones.

## Age-associated extramedullary changes accelerate myeloid-biased hematopoiesis

We found that the fluctuating LT-HSC/ST-HSC ratio in the bulk-HSC compartment corresponded with the myeloid-biased hematopoiesis associated with aging. In contrast, the degree of myeloid bias observed in mice older than 23 months without transplantation (*Figure 1A*; 49.0 ± 23.4%) was significantly greater than in recipient mice receiving young mixed HSCs with a 5:5 ratio 16 weeks after transplantation (*Figure 6D*; 30.8 ± 30.8%). This difference indicated that other intrinsic or extrinsic factors might exist in mice older than 23 months to promote myeloid-biased hematopoiesis. To investigate this further, we transplanted 10 GFP$^+$ young LT-HSCs into young or 2-year-old recipient mice and examined their PB (*Figure 7A*). We limited the observation period to 12 weeks because many aged recipient mice had died prior to that point (*Figure 7B*). PB analysis revealed that aged recipient mice receiving young LT-HSCs produced significantly more myeloid cells than young recipient mice (*Figure 7C–E*). These results suggested that recipient-dependent extrinsic factors, rather than intrinsic factors of transplanted HSCs, had a greater impact on hematopoietic cell differentiation. Then, we examined BM to determine what extrinsic factors affected differentiation of transplanted HSCs. We performed BM analysis for the mice denoted by † in *Figure 7C* because many of the aged mice had died before the analysis. Percentages of CMP and GMP, downstream components of myeloid differentiation, were stable or significantly lower in aged recipient mice than in young recipient mice, just as they were in non-treated aged mice (*Figures 1B and 7F*). In addition, the percentage of common lymphoid progenitor (CLP), a downstream component of lymphoid differentiation, did not differ between young and aged recipients (*Figure 7F*). In contrast to previous studies (*Pinho et al., 2018*; *Ergen et al., 2012*), these findings suggested that intramedullary age-associated changes may not have a significant impact on LT-HSC differentiation.

Thymi and spleens of recipient mice were then examined, as these are the structures of lymphoid maturation and production of lymphoid progenitors following migration from BM and prior to their appearance in PB. The thymus and spleen analyses were also performed on the mice denoted by † in *Figure 7C*. Donor cells were not microscopically detectable in the majority of aged recipient mice (*Figure 7G, I*), and quantitative analysis using a flow cytometer revealed that the frequency of donor cells in spleens and thymi of aged recipients was significantly lower than in young recipients (*Figure 7H, J*). These results indicate that the process of lymphoid lineage differentiation is impaired in the spleens and thymi of aged mice compared to young mice, or that differentiating cells in the BM do not successfully migrate into these secondary lymphoid organs. These factors contribute to the enhanced myeloid-biased hematopoiesis in PB due to a decrease in de novo lymphocyte production.

## Discussion

Age-related, myeloid-biased hematopoiesis has been suggested as a potential reason for the decline in acquired immunity in the elderly (*Geiger et al., 2013*). This process has been linked to an increase in myeloid-biased HSC clones in the bulk-HSC fraction identified by surface antigens through aging. By isolating LT-HSCs and ST-HSCs, respectively, we discovered that age-related changes in the ratio of ST-HSCs to LT-HSCs in bulk-HSCs are responsible for myeloid-biased hematopoiesis.

When aged bulk-HSCs were transplanted into young mice, post-transplant hematopoiesis in recipient mice was significantly biased toward myeloid in PB compared to transplantation of young bulk-HSCs (*Beerman et al., 2010*; *Dykstra et al., 2011*; *Sudo et al., 2000*). Historically, this myeloid-biased hematopoiesis derived from aged bulk-HSCs has been considered as evidence supporting expansion of myeloid-biased HSC clones with aging (*Geiger et al., 2013*; *Haan and Lazare, 2018*). Other studies suggest that blockage of lymphoid hematopoiesis in aged mice results in myeloid-skewed hematopoiesis through alternative mechanisms. However, this result should be interpreted carefully, since Busulfan was used for myeloablative treatment in this study (*Montecino-Rodriguez et al., 2019*). To clarify the cause for this disparity, we chose to examine the LT-HSC fraction, which persists in the BM for long periods of time and is enriched for aging-related alterations. We discovered that post-transplant hematopoiesis of aged LT-HSCs did not vary from that of young LT-HSCs in

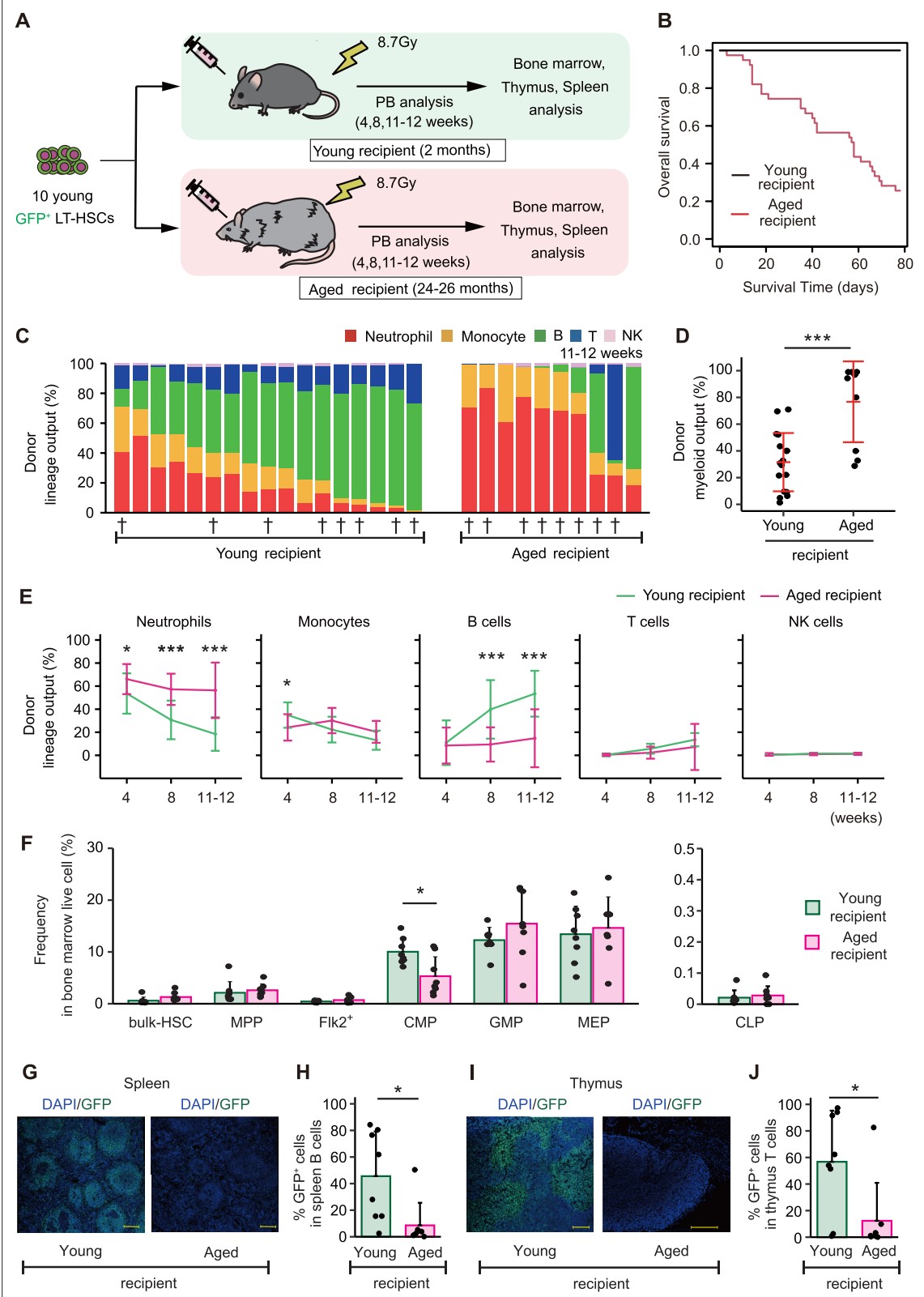

**Figure 7.** Age-associated physiological changes drive differentiation of long-term hematopoietic stem cells (LT-HSCs) toward myeloid cells. (**A**) Experimental design for assessing the impact of age-associated physiological changes on differentiation of LT-HSCs. Ten GFP+ LT-HSCs sorted from young (2–3 months) *Hoxb5*-tri-mCherry GFP mice were transplanted with $2 \times 10^5$ CD45.1+/CD45.2+ supporting cells into lethally irradiated young or aged recipient mice. We defined donor cells as GFP+ cells and supporting cells as CD45.1+/CD45.2+ cells. (**B**) Survival rate of recipient

*Figure 7 continued on next page*

*Figure 7 continued*

mice in each group. (**C**) Donor lineage output in young or aged recipient mice 11–12 weeks after transplanting young LT-HSCs (young recipient, *n* = 17; aged recipient, *n* = 10). (**D**) Myeloid output (frequency of donor myeloid cells in donor fraction) in young or aged recipient mice 11–12 weeks after transplantation. (**E**) Kinetics of lineage output from donor LT-HSCs in young or aged recipient mice 4, 8, 11, and 12 weeks after transplantation. (**F**) Average frequency of donor bulk-hematopoietic stem cell (HSC) and progenitor cells in donor bone marrow (BM) live cells (young recipient, *n* = 8; aged recipient, *n* = 8). BM samples were taken from mice denoted by † in (**C**). (**G**) Representative immunofluorescence images of frozen spleen sections derived from young or aged recipient mice. Green: donor cells (GFP fluorescence); blue: DNA (4',6-diamidino-2-phenylindole, DAPI); scale bar: 200 µm. (**H**) Frequency of donor cells in spleen B cells of young or aged recipient mice (young recipient, *n* = 8; aged recipient, *n* = 8). Spleens are taken from mice denoted by † in (**C**). (**I**) Representative immunofluorescence images of frozen thymus sections derived from young or aged recipient mice. Green: donor cells (GFP fluorescence); blue: DNA (DAPI); scale bar: 200 µm. (**J**) Frequency of donor cells in thymus T cells of young or aged recipient mice (young recipient, *n* = 8; aged recipient, *n* = 8). Thymi are taken from mice denoted by † in (**C**). *$p < 0.05$, ***$p < 0.001$. Error bars represent standard deviation. Data represent two independent experiments.

terms of differentiation capacity. Expression of myeloid-related genes was not significantly altered when LT-HSCs from young and aged mice were compared. Therefore, we inferred that age-related myeloid-biased hematopoiesis cannot be attributed to an increase in myeloid-biased HSCs, at least among LT-HSCs. Given that bulk-HSCs consist of LT-HSCs and ST-HSCs, and that LT-HSCs exhibit no change in differentiation potential with age, we hypothesized that ST-HSCs may be responsible for the myeloid-skewing phenotype in PB. Transcriptomic analysis revealed no enrichment of myeloid-related genes in the ST-HSC fraction, ruling out the possibility of an increase in myeloid-biased HSCs in the ST-HSC fraction. Alternatively, we observed a proportionate decrease in ST-HSCs in bulk-HSCs accompanying an increase in LT-HSCs. We postulated that this relative decrease in ST-HSC ratio is responsible for myeloid-biased hematopoiesis. When isolated LT-HSCs and ST-HSCs from young mice were reconstituted to replicate an aged BM type in terms of the LT-HSC/ST-HSC ratio, we discovered that PB was strongly skewed toward myeloid cells after transplantation of reconstituted HSCs. In contrast, transplantation of reconstituted HSCs with a young BM type resulted in a significantly lymphoid-skewed PB profile.

On the other hand, it has been reported that myeloid-biased HSCs have a more persistent capacity for self-renewal (*Muller-Sieburg et al., 2004*). In fact, in our prior investigation comparing PB 16 weeks post-transplant, myeloid cell production in PB was greater when LT-HSCs were transplanted, while ST-HSC transplantation resulted in a considerable increase in lymphoid cell production (*Chen et al., 2016*). It is therefore possible that the LT-HSCs and ST-HSCs we isolated utilizing the Hoxb5 reporter system are myeloid- and lymphoid-biased HSCs, respectively. However, the myeloid-producing capacity of LT-HSCs and ST-HSCs 4 weeks post-transplantation is equivalent (*Chen et al., 2016*). In contrast, 16 weeks after transplantation, ST-HSC transplant recipients demonstrate negligible myeloid development and a predominance of lymphocytes, unlike LT-HSC transplant recipients (*Chen et al., 2016*). These characteristics show that early post-transplant ST-HSCs lack the phenotypic of lymphoid-biased HSCs. When ST-HSC-transplanted BM cells are employed as donors for secondary transplantation, no donor-cell-derived hematopoiesis is detected in recipients (*Chen et al., 2016*). In addition, HSCs are not observed in the BM following transplantation of ST-HSCs (*Figure 5*). These results indicate that ST-HSCs lose their capacity for self-renewal earlier after transplantation than LT-HSCs. In addition, phenotypic examination of lymphocytes in PB after ST-HSC transplantation revealed that nearly all of them were memory cells, and that there was no de novo supply of T cells. In other words, ST-HSCs may have been erroneously classified as lymphoid-biased HSCs due to the preservation of memory lymphocytes with a long half-life in the peripheral circulation, despite the absence of de novo hematopoiesis following transplantation of ST-HSCs. These findings suggest that the bias toward myeloid and lymphoid lineages in post-transplant PB is not regulated by heterogeneity of multipotency, but by heterogeneity of self-renewal capacity.

Multiple extrinsic factors have been implicated as HSC-independent causes of myeloid-biased hematopoiesis associated with aging. Some findings suggest that cell interaction with niche cells in BM is altered (*Pinho et al., 2018*), whereas others claim that chronic inflammation in aged mice lowers production of lymphoid progenitors and induces myeloid-biased hematopoiesis (*Montecino-Rodriguez et al., 2019*; *Ergen et al., 2012*). To identify HSC-independent causes of myeloid-biased hematopoiesis specifically, we transplanted young LT-HSCs into young or aged mice. Cellular differentiation observed in BM revealed that proportions of GMP, MEP, and CLP are unchanged between young and aged recipients. In contrast, an examination of spleens and thymi, the sites of lymphocyte

maturation, revealed the absence of donor cells in aged recipients. These findings suggest that myeloid cells increase in aging mice due to a relative decrease in the ratio of mature lymphocytes in PB, resulting from inhibition of lymphocyte maturation outside the BM, such as in the thymus and spleen, rather than a significant change in cell differentiation caused by the intramedullary environment. Attempts have been made in the past to restore thymus function in aged mice by administering keratinocyte growth factor (*Rossi et al., 2007b*). Indeed, in aged mice, lymphocyte hematopoiesis recovers, which is consistent with our results that the extramedullary environment influences the myeloid-bias phenotype in PB (*Rossi et al., 2007b*).

It has been clinically documented that recovery of the lymphocyte fraction in PB is delayed in elderly donors, compared to young donors (*González-Vicent et al., 2017*; *Baron et al., 2006*). Although it is undeniable that this phenomenon may be due to decreased HSC hematopoietic potential, it may also be due to a change in the ratio of LT-HSC/ST-HSCs in bulk-HSCs, as described in our report, which may give the appearance of a myeloid bias in PB after transplantation. It has also been noted that lymphocyte recovery is delayed in elderly patients compared to younger patients (*Berger et al., 2008*; *van der Maas et al., 2021*). Age-related loss in thymus and spleen functions has been recorded in humans, implying that the same mechanism is acting as in our mouse observations. If it is possible to assess and regulate the LT-HSC/ST-HSC ratio in donors and thymus/spleen function in recipients in the future, it will help to develop accurate models for predicting lymphocyte recovery after transplantation as well as new transplantation strategies.

In contrast, our findings should be considered in light of some limitations. In this report, we primarily performed 10–20 cell transplantation assays. Therefore, the current theory should be revalidated using single-cell technology with lineage tracing system (*Yamamoto et al., 2018*; *Rodriguez-Fraticelli et al., 2020*). This approach will investigate changes in the self-renewal capacity of individual HSCs and their subsequent differentiation into progenitor cells and PB cells. In the co-transplantation assay shown in *Figure 3*, the myeloid lineage output derived from young and aged LT-HSCs was comparable (young LT-HSC: 51.4 ± 31.5% vs. aged LT-HSC: 47.4 ± 39.0%, p = 0.82). Although no significant difference was detected, the small sample size (*n* = 8) may limit the sensitivity of the assay to detect subtle myeloid-biased phenotypes. Additionally, in this study, we purified LT-HSCs using the Hoxb5 reporter system and employed a moderate conditioning regimen (8.7 Gy). To have a better picture of total donor contribution, total PB chimerism is presented in *Figure 8—figure supplement 1* and we cannot exclude the possibility that these factors may have influenced the results. Therefore, it would be ideal to validate our findings using alternative LT-HSC markers and different conditioning regimens.

In summary, we have demonstrated for the first time that the ratio of LT-HSC/ST-HSC is important in age-related myeloid-skewed hematopoiesis as an HSC-dependent factor and that aging-related thymus and spleen dysfunction also contributes significantly as an HSC-independent factor. Furthermore, here we propose a 'self-renewal heterogeneity model' as a new mechanism for hematopoietic heterogeneity, including myeloid-biased hematopoiesis in aged mice (*Figure 8*). In addition to this report, we have also shown that cellular differentiation in the PB after transplantation can vary by changing transplantation conditions, ultimately altering self-renewal ability (*Sakamaki et al., 2021*; *Nishi et al., 2022*). This highlights the significance of deciphering molecular processes that regulate the heterogeneity of HSC self-renewal capacity, which will help to provide a better understanding of the hematopoietic system and the hematopoietic hierarchy in the future.

## Methods
### Mice

Mice with an *Hoxb5*-tri-mCherry (C57BL/6J background), derived from our previous work (*Chen et al., 2016*), were harvested for donor cells for transplantation, PB, and BM analysis. CAG-EGFP mice (C57BL/6J background) were bred with *Hoxb5*-tri-mCherry for transplantation experiments. Eight- to fourteen-week-old C57BL/6-Ly5.1 mice (purchased from Sankyo Labo Service) were used as recipients for transplantation assays. For aged recipients, 2-year-old CD45.2 C57BL/6J mice were purchased from CLEA Japan. Supporting BM cells were collected from 8- to 12-week-old C57BL/6-Ly5.1 × C57BL/6J (F$_1$ mice CD45.1$^+$/CD45.2$^+$). All mice were housed in specific pathogen-free conditions and were carefully observed by staff members. Mice were bred according to RIKEN or Kyoto University.

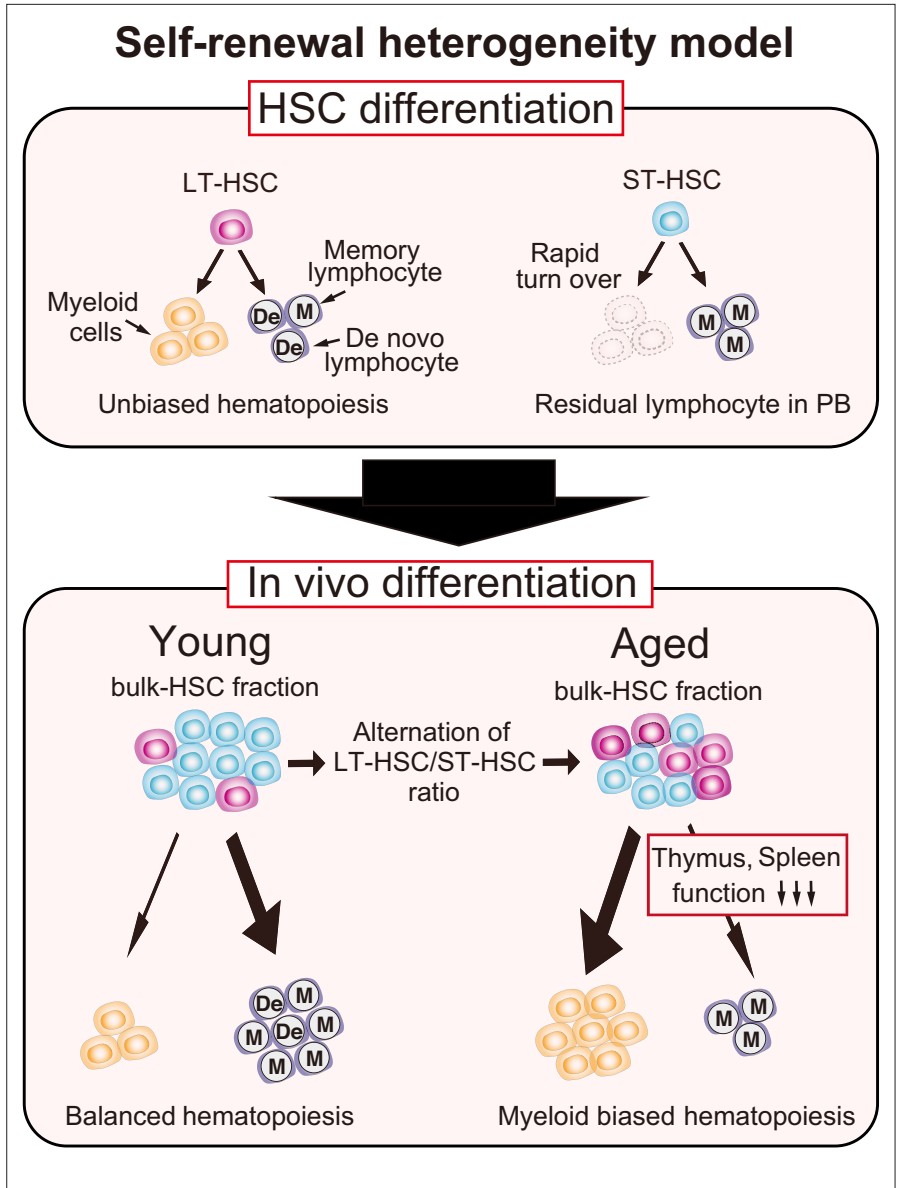

**Figure 8.** Our new model: self-renewal heterogeneity model. It has been thought that there were myeloid (My-) or lymphoid biased (Ly-) hematopoietic stem cells (HSCs), and that clonal selection of My-HSCs caused age-associated myeloid-biased hematopoiesis. However, in our model, long-term hematopoietic stem cells (LT-HSCs) represent unbiased hematopoiesis throughout life. Short-term hematopoietic stem cells (ST-HSCs) lose their hematopoietic ability within a short period, and memory-type lymphocytes remain in the peripheral blood (PB) after ST-HSC transplantation. These remaining memory-type lymphocytes make it look as if ST-HSCs are lymphoid-biased (the upper section). As a result, the age-associated relative decrease of ST-HSCs in bulk-HSC fraction causes myeloid-biased hematopoiesis with age. Additionally, the blockage of lymphoid differentiation at the spleen and thymus accelerates further myeloid-biased hematopoiesis in aged mice (the lower section).

The online version of this article includes the following figure supplement(s) for figure 8:

**Figure supplement 1.** Analysis of donor chimerism in whole blood cells for each transplantation assay.

All animal protocols were approved by the RIKEN Center for Biosystems Dynamics Research (A2017-07-10) and Kyoto University (MedKyo 19033, MedKyo 20004).

## Flow cytometry, cell sorting, and BM analyses

Flow cytometry and cell sorting were performed on a FACS Aria II cell sorter (BD Biosciences) and analyzed using FlowJo software (BD Biosciences). BM cells were collected from bilateral tibias, femurs, humeri, and pelvises in $Ca^{2+}$- and $Mg^{2+}$-free PBS supplemented with 2% heat-inactivated bovine serum (Gibco) and 2 mM EDTA (Thermo Fisher Scientific). Cells were passed through 100 and 40 µm strainers (Corning) before analysis and sorting. Prior to staining, samples were blocked with 50 µg/ml rat IgG (Sigma-Aldrich) for 15 min. To enrich HSCs and progenitor populations for sorting and BM analyses, cells were stained with APC-eFluor780-conjugated anti-c-Kit (clone: 2B8) and fractionated using anti-APC magnetic beads and LS columns (both Miltenyi Biotec). c-Kit$^+$ cells were then stained with combinations of antibodies as described in *Supplementary file 1*. All staining was incubated at 4°C for 30 min, except for CD34 staining, which was incubated for 90 min. Samples were washed twice after staining. Prior to flow cytometry or cell sorting, samples were stained with SYTOX Red Dead Cell Stain (Life Technologies) or 7-aminoactinomycin D (BioLegend). Cells were double sorted for purity. For the combined analysis of the upstream (HSC, MPP, and Flk2$^+$) and downstream (CLP, MEP, CMP, and GMP) fractions in *Figure 1B*, we normalized by c-Kit$^+$ cells because we performed a c-Kit enrichment for the BM analysis.

## Transplantation and PB analyses

Twelve to twenty-four hours prior to transplantation, C57BL/6-Ly5.1 mice, or aged C57BL/6J recipient mice were lethally irradiated with single doses of 8.7 Gy. For transplantation assays, donor cells were first combined with $2 \times 10^5$ whole BM supporting cells (C57BL/6-Ly5.1 × C57BL/6 J $_F$1 mice CD45.1$^+$/CD45.2$^+$) in 200 µl of PBS with 2% FBS, and then injected into the retro-orbital venous plexus. For evaluation of post-transplant kinetics, PB was collected and analyzed. At each time point, 50 µl of blood were collected from the tail vein and re-suspended in $Ca^{2+}$- and $Mg^{2+}$-free PBS supplemented with 2 mM EDTA. Red blood cells were lysed twice on ice for 3 min with BD Pharm Lyse Buffer (BD Pharmingen). Identification of leukocyte subsets was performed by staining with antibodies. Antibody information is described in *Supplementary file 1*. For evaluation of lineage output, the frequency of each lineage (NK cell, B cell, T cell, neutrophil, and monocyte) was determined in the whole fraction. The analysis of donor lineage output was restricted to donor cells showing ≥0.1% donor chimerism at the last PB analysis to allow reliable detection. The percentage of donor chimerism in PB was defined as the percentage of CD45.1$^-$CD45.2$^+$ cells among total CD45.1$^-$CD45.2$^+$ and CD45.1$^+$CD45.2$^+$ cells. PB data represent mice which survived until the last PB analysis.

## Thymus and spleen analysis

Thymi and spleens were harvested and disrupted into a single-cell suspension. Cells were passed through 100 and 40 µm strainers (Corning). Isolated thymocytes and splenocytes were incubated on ice for 30 min with appropriately diluted antibodies in staining buffer. Identification of B cells in spleens and T cells in thymi was performed by staining with antibodies described in *Supplementary file 1*. Prior to flow cytometry or cell sorting, samples were stained with SYTOX Red Dead Cell Stain (Life Technologies).

## Tissue imaging

Freshly dissected spleens and thymi were fixed in 4% PFA (Nacalai tesque) in PBS for 24 hr at 4°C. After PFA washout with PBS, tissues were cryoprotected with 30% sucrose in PBS for 24 hr at 4°C, embedded in Tissue-Tek O.C.T. compound (Sakura Finetek), and snap-frozen in liquid nitrogen. Serial 10 µm longitudinal cryostat sections were obtained using CryoStar NX50 (Thermo Fisher Scientific). Cell nuclei were counterstained with 4',6-diamidino-2-phenylindole (1 µg/ml, Roche). To reduce autofluorescence, tissue sections were treated with Vector TrueVIEW autofluorescence quenching kit (Vector Laboratories). Images were obtained from Leica TCS SP8 (Leica).

## RNA sequencing

Total RNA was isolated with Trizol (Thermo Fisher Scientific) and cleaned up using RNeasy MinElute columns (QIAGEN). cDNA libraries were prepared from bulk-HSCs, ST-HSCs, and LT-HSCs using a KAPA RNA HyperPrep kit with RiboErase (HMR) (Kapa Biosystems) and sequenced using a Hiseq 1500 (Illumina) to obtain 2 × 127 basepair (bp) paired-end reads. Three replicates were sequenced for each population. Raw transcriptome sequence data were mapped to the genome (mm10) using HISAT2 (ver 2.1.0) (*Kim et al., 2015*). Alignments were then passed to StringTie (ver 1.3.4d), which was used to assemble and quantify transcripts in each sample (*Pertea et al., 2016*). EdgeR (ver 3.22.5) was then used to compare all transcripts across conditions and to produce tables and plots of differentially expressed genes and transcripts (*Robinson et al., 2010*). Normalization with the TMM method was performed with the edgeR package in Bioconductor (https://bioconductor.org/). Genes (12,808 genes) were selected for hierarchical clustering based on the mean of TMM normalized read counts across all samples' cells (mean ≥ 1). Then, clustering was performed by using the hclust function for R (distance = 'Spearman's correlation'; method = 'ward.D2'). We depicted Venn diagrams and performed GSEA analyses in *Figure 4D–G* and *Figure 4—figure supplement 1C–D* using previously published gene sets after excluding genes that could not be annotated by our transcriptome dataset. GSEA was performed using GSEA software (http://www.broadinstitute.org/gsea) with default settings (*Subramanian et al., 2005*).

## Quantification and statistical analyses

Statistical analyses were performed using ggplot2 in R (version 4.1.2) or Microsoft Excel. Sample size for each experiment and replicate numbers of experiments are included in figure legends. Statistical significance was determined using Welch's $t$ test. p values <0.05 were considered significant.

## Acknowledgements

We gratefully acknowledge Hiroshi Kiyonari for animal care and providing recipient mice at RIKEN BDR. Shigehiro Kuraku, Mitsutaka Kadota, and Chiharu Tanegashima provided technical support for RNA sequencing and analysis at RIKEN BDR. Momo Fujii performed imaging experiments at RIKEN BDR. Hitomi Oga, Kayoko Nagasaka, and Masaki Miyahashi provided laboratory management at Kobe University. The authors gratefully acknowledge ongoing support for this work: Masanori Miyanishi was supported by the Japan Society for the Promotion of Science (JSPS) KAKENHI Grant Numbers JP17K07407 and JP20H03268, The Mochida Memorial Foundation for Medical and Pharmaceutical Research, The Life Science Foundation of Japan, The Takeda Science Foundation, The Astellas Foundation for Research on Metabolic Disorders, AMED-PRIME, AMED under Grant Number JP18gm6110020. Taro Sakamaki was supported by the JSPS Core-to-Core Program, JSPS KAKENHI Grant Numbers JP21K20669 and JP22K16334, RIKEN Junior Research Associate Program. Katsuyuki Nishi was supported by JSPS Grant Numbers KAKENHI JP18J13408 and 23K15326.

## Additional information

### Funding

| Funder | Grant reference number | Author |
| --- | --- | --- |
| Japan Society for the Promotion of Science | JP17K07407 | Masanori Miyanishi |
| Mochida Memorial Foundation for Medical and Pharmaceutical Research | | Masanori Miyanishi |
| Life Science Foundation of Japan | | Masanori Miyanishi |
| Takeda Science Foundation | | Masanori Miyanishi |

| Funder | Grant reference number | Author |
| --- | --- | --- |
| Astellas Foundation for Research on Metabolic Disorders | | Masanori Miyanishi |
| Japan Agency for Medical Research and Development | JP18gm6110020 | Masanori Miyanishi |
| Japan Society for the Promotion of Science | JP21K20669 | Taro Sakamaki |
| Japan Society for the Promotion of Science | 23K15326 | Katsuyuki Nishi |
| RIKEN | Junior Research Associate Program | Taro Sakamaki |
| Japan Society for the Promotion of Science | JP20H03268 | Masanori Miyanishi |
| Japan Society for the Promotion of Science | The JSPS Core to Core program | Taro Sakamaki |
| Japan Society for the Promotion of Science | JP22K16334 | Taro Sakamaki |
| Japan Society for the Promotion of Science | JP18J13408 | Katsuyuki Nishi |

The funders had no role in study design, data collection, and interpretation, or the decision to submit the work for publication.

#### Author contributions
Katsuyuki Nishi, Conceptualization, Data curation, Formal analysis, Funding acquisition, Validation, Investigation, Visualization, Methodology, Writing – original draft, Writing – review and editing; Taro Sakamaki, Formal analysis, Investigation, Writing – review and editing; Akiomi Nagasaka, Kevin Shuolong Kao, Nobuyuki Yamamoto, Writing – review and editing; Kay Sadaoka, Investigation, Writing – review and editing; Masahide Asano, Resources, Writing – review and editing; Akifumi Takaori-Kondo, Supervision, Writing – review and editing; Masanori Miyanishi, Conceptualization, Resources, Formal analysis, Supervision, Funding acquisition, Project administration, Writing – review and editing

#### Author ORCIDs
Katsuyuki Nishi ⓘ https://orcid.org/0009-0009-4296-7619
Nobuyuki Yamamoto ⓘ https://orcid.org/0000-0002-1006-7394
Akifumi Takaori-Kondo ⓘ https://orcid.org/0000-0001-7678-4284
Masanori Miyanishi ⓘ https://orcid.org/0000-0002-0527-3652

#### Ethics
All animal protocols were approved by the RIKEN Center for Biosystems Dynamics Research (A2017-07-10) and Kyoto University (MedKyo 19033, MedKyo 20004).

Reviewer #1 (Public review): https://doi.org/10.7554/eLife.95880.4.sa1
Reviewer #2 (Public review): https://doi.org/10.7554/eLife.95880.4.sa2
Reviewer #3 (Public review): https://doi.org/10.7554/eLife.95880.4.sa3
Author response https://doi.org/10.7554/eLife.95880.4.sa4

## Additional files

### Supplementary files
MDAR checklist

Supplementary file 1. List of used antibodies.

## Data availability

Sequencing data has been deposited in the Gene Expression Omnibus under accession code GSE226803. Correspondence and requests for materials should be addressed to MM (miya75@med. kobe-u.ac.jp).

The following dataset was generated:

| Author(s) | Year | Dataset title | Dataset URL | Database and Identifier |
|---|---|---|---|---|
| Nishi K, Miyanishi M | 2023 | Transcriptome analysis of bulk-HSCs, LT-HSCs and ST-HSCs | https://www.ncbi. nlm.nih.gov/geo/ query/acc.cgi?acc= GSE226803 | NCBI Gene Expression Omnibus, GSE226803 |

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
