## [Editor Report · eLife Assessment]

This manuscript provides **valuable** insights into the heterogeneity of hematopoietic stem cells and age-associated myeloid-biased hematopoiesis. While several aspects of the study are intriguing and merit further investigation, the current results remain **incomplete** and additional data are necessary to substantiate the conclusions. Some of the methods and data analyses partially support the claims.

---

## [Referee Report · Reviewer #1 (Public review)]

In this study, Nishi et al. claim that the ratio of long-term hematopoietic stem cell (LT-HSC) versus short-term HSC (ST-HSC) determines the lineage output of HSCs and reduced ratio of ST-HSC in aged mice causes myeloid-biased hematopoiesis. Authors used Hoxb5 reporter mice to isolated LT-HSC and ST-HSC and performed molecular analyses and transplantation assays to support their arguments. How hematopoietic system becomes myeloid-biased upon aging is an important question with many implications in disease context as well. However, this study needs more definitive data.

(1) Authors' experimental designs have some caveats to definitely support their claims. Authors claimed that aged LT-HSCs have no myeloid-biased clone expansion using transplantation assays. In these experiments, authors used 10 HSCs and young mice as recipients. Given the huge expansion of old HSC by number and known heterogeneity in immunophenotypically defined HSC populations, it is questionable how 10 out of so many old HSCs (an average of 300,000 up to 500,000 cells per mouse; Mitchell et al., Nature Cell Biology, 2023) can faithfully represent old HSC population. The Hoxb5+ old HSC primary and secondary recipient mice data (Fig. 2C and D) support this concern. In addition, they only used young recipients. Considering the importance of inflammatory aged niche in the myeloid-biased lineage output, transplanting young vs old LT-HSCs into aged mice will complete the whole picture.

In response to the above comments, the authors calculated the required sample size as approximately 384 cells to represent 500,000 HSCs per old mouse. Based on the total 1260 cells used throughout the whole manuscript (Figures 2, 3, 5, 6, S3, and S6), the authors claimed that the data is reflecting old HSC behavior. However, 384 cells represent HSCs from one old mouse. Following the authors' logic, they did only 3.2 mice (1260/384) experiment for the whole manuscript to make their argument. N of 3 is not enough, especially for old mice experiments considering the heterogeneity of aged mice. Also, they did not address the comment regarding inflammatory aged niche effects.

(2) Authors' molecular data analyses need more rigor with unbiased approaches. They claimed that neither aged LT-HSCs nor aged ST-HSCs exhibited myeloid or lymphoid gene set enrichment but aged bulk HSCs, which are just a sum of LT-HSCs and ST-HSCs by their gating scheme (Fig. 4A), showed the "tendency" of enrichment of myeloid-related genes based on the selected gene set (Fig. 4D). Although the proportion of ST-HSCs is reduced in bulk HSCs upon aging, since ST-HSCs do not exhibit lymphoid gene set enrichment based on their data, it is hard to understand how aged bulk HSCs have more myeloid gene set enrichment compared to young bulk HSCs. This bulk HSC data rather suggest that there could be a trend toward certain lineage bias (although not significant) in aged LT-HSCs or ST-HSCs. Authors need to verify the molecular lineage priming of LT-HSCs and ST-HSCs using another comprehensive dataset.

(3) Although authors could not find any molecular evidence for myeloid-biased hematopoiesis from old HSCs (either LT or ST), they argued that the ratio between LT-HSC and ST-HSC causes myeloid-biased hematopoiesis upon aging based on young HSC experiments (Fig. 6). However, old ST-HSC functional data showed that they barely contribute to blood production unlike young Hoxb5- HSCs (ST-HSC) in the transplantation setting (Fig. 2). Is there any evidence that in unperturbed native old hematopoiesis, old Hoxb5- HSCs (ST-HSC) still contribute to blood production? To answer this question, authors performed additional experiments with increased cell number (Fig. S6). Although Fig. S6.D data has a statistical significance, it is questionable how biologically meaningful it is. More fundamental question is back to the representability. Can this cell number used in this experiment represent old HSC (either LT or ST) behavior?

---

## [Referee Report · Reviewer #2 (Public review)]

Summary:

Nishi et al, investigate the well-known and previously described phenomenon of age-associated myeloid-biased hematopoiesis. Using a previously established HoxB5mCherry mouse model, they used HoxB5+ and HoxB5- HSCs to discriminate cells with long-term (LT-HSCs) and short-term (ST-HSCs) reconstitution potential and compared these populations to immunophenotypically defined 'bulk HSCs' that consists of a mixture of LT-HSC and ST-HSCs. They then isolated these HSC populations from young and aged mice to test their function and myeloid bias in non-competitive and competitive transplants into young and aged recipients. Based on quantification of hematopoietic cell frequencies in the bone marrow, peripheral blood, and in some experiments the spleen and thymus, the authors argue against the currently held belief that myeloid-biased HSCs expand with age.

While aspects of their work are fascinating and might have merit, several issues weaken the overall strength of the arguments and interpretation. Multiple experiments were done with a very low number of recipient mice, showed very large standard deviations, and had no statistically detectable difference between experimental groups. While the authors conclude that these experimental groups are not different, the displayed results seem too variable to conclude anything with certainty. The sensitivity of the performed experiments (e.g. Fig 3; Fig 6C, D) is too low to detect even reasonably strong differences between experimental groups and is thus inadequate to support the author's claims. This weakness of the study is not acknowledged in the text and is also not discussed. To support their conclusions the authors need to provide higher n-numbers and provide a detailed power analysis of the transplants in the methods section.

As the authors attempt to challenge the current model of the age-associated expansion of myeloid-biased HSCs (which has been observed and reproduced by many different groups), ideally additional strong evidence in the form of single-cell transplants is provided.

It is also unclear why the authors believe that the observed reduction of ST-HSCs relative to LT-HSCs explains the myeloid-biased phenotype observed in the peripheral blood. This point seems counterintuitive and requires further explanation.

Based on my understanding of the presented data, the authors argue that myeloid-biased HSCs do not exist, as:

(a) they detect no difference between young/aged HSCs after transplant (mind low n-numbers and large std);

(b) myeloid progenitors downstream of HSCs only show minor or no changes in frequency and (c) aged LT-HSCs do not outperform young LT-HSC in myeloid output LT-HScs in competitive transplants (mind low n-numbers and large std!!!).

However, given the low n-numbers and high variance of the results, the argument seems weak and the presented data does not support the claims sufficiently. That the number of downstream progenitors does not change could be explained by other mechanisms, for instance, the frequently reported differentiation short-cuts of HSCs and/or changes in the microenvironment.

Strengths:

The authors present an interesting observation and offer an alternative explanation of the origins of aged-associated myeloid-biased hematopoiesis. Their data regarding the role of the microenvironment in the spleen and thymus appears to be convincing.

Weaknesses:

"Then, we found that the myeloid lineage proportions from young and aged LT-HSCs were nearly comparable during the observation period after transplantation (Fig. 3, B and C)."

[Comment to the authors]: Given the large standard deviation and low n-numbers, the power of the analysis to detect differences between experimental groups is very low. Experimental groups with too large standard deviations (as displayed here) are difficult to interpret and might be inconclusive. The absence of clearly detectable differences between young and aged transplanted HSCs could thus simply be a false-negative result. The shown experimental results hence do not provide strong evidence for the author's interpretation of the data. The authors should add additional transplants and include a detailed power analysis to be able to detect differences between experimental groups with reasonable sensitivity.

Line 293: "Based on these findings, we concluded that myeloid-biased hematopoiesis observed following transplantation of aged HSCs was caused by a relative decrease in ST-HSC in the bulk-HSC compartment in aged mice rather than the selective expansion of myeloid-biased HSC clones."

[Comment to the authors]: Couldn't that also be explained by an increase in myeloid-biased HSCs, as repeatedly reported and seen in the expansion of CD150+ HSCs? It is not intuitively clear why a reduction of ST-HSCs clones would lead to a myeloid bias. The author should try to explain more clearly where they believe the increased number of myeloid cells comes from. What is the source of myeloid cells if the authors believe they are not derived from the expanded population of myeloid-biased HSCs?

New comment for the authors:

While the authors provide new evidence, clarify the text, and adjust their interpretation, the presented data remain weak and do not convincingly challenge the current paradigm. As myeloid-biased HSC expansion with age has been observed and published by many different groups, the authors need to provide much stronger evidence to challenge the observations of others. Key experiments that might support their claims had been suggested, but as indicated, the authors plan to provide these much more rigorous experiments in future studies. As it stands, the overall conclusions of this manuscript thus remain weak and preliminary.

In an attempt to quantify the absolute cell number of HSPC subpopulations, the authors use a usual readout and quantify "Number of cells per minute of analysis time". This appears to be a quick and dirty reanalysis of already existing flow cytometry data. Unfortunately, this quantification cannot count the absolute number of cells reliably, as the number of cells per minute recorded is heavily influenced by the abundance of other cell populations. Instead, the author should have counted the absolute number of HSCs, MPPs, GMPs, etc. per femur, which is typically done to address this question.

At this point, as authors are seemingly not willing to provide additional hard evidence to support their claims in this study and are instead in the process of preparing additional data for a future manuscript, I believe this study, as it stands (although weak), suggests an interesting alternative model. Despite being highly controversial, this alternative model warrants future investigations and discussions in the field. As always, it will also be important to reproduce these findings independently in other labs. As my concerns and the concerns of the other reviewers are documented and available to read by others, I believe the manuscript should be published in its current form to stimulate critical discussion and future investigations of the current model.

---

## [Referee Report · Reviewer #3 (Public review)]

In this manuscript, Nishi et al. propose a new model to explain the previously reported myeloid-biased hematopoiesis associated with aging. Traditionally, this phenotype has been explained by the expansion of myeloid-biased hematopoietic stem cell (HSC) clones during aging. Here, the authors question this idea and show how their Hoxb5 reporter model can discriminate long-term (LT) and short-term (ST) HSC and characterized their lineage output after transplant. From these analyses, the authors conclude that changes during aging in the LT/ST HSC proportion explain the myeloid bias observed.

Comments on revisions:

I appreciate the authors' reply to some of my comments. However, there are some key aspects that remain unresolved. Please see below.

- The authors propose a critical change in the way we consider the mechanisms leading to lineage biased hematopoiesis during aging. As Reviewer 2 mentioned, such a strong claim needs to be supported by solid experimental data. Unfortunately, the level of variability in key in vivo experiments (Figure 2 and 3) diminishes the robustness of these results.

The authors argue that even with the low number of mice used in some of these experiments and the high level of variability, differences still reach (or not) statistical significance according to their analysis. I am not an expert on statistics but the only test that is mentioned is their methodology is a Welch's t test, which is only appropriate for data following a normal distribution. A more rigorous statistical analysis should be performed to sustain the claims included in the current manuscript.

- The chosen irradiation regiment might contribute to the uncertainty of the data and influence their interpretation. As the authors show in their response to my "comment to our #3-4 response", there is a considerable (and variable) amount of "radioresistant" CD45.1+CD45.2- cells in their primary recipients, which become concerningly high in the secondary transplant. This is not found in previous publications focused on this topic and, therefore, it makes it difficult to compare those studies with the present manuscript. The inclusion of this aspect in the text is appreciated but definitely reduces the impact of their claims.

- The correction introduced in the main text as an answer to the original comment #3-6 is still misleading. There is an assumption for GMP, CMP and MEP to increase with age if myeloid-biased HSC clones increase with age ("in contrast to what we anticipated"). Again, the link between these two changes could be more complex than just a direct correlation.

---

## [Author Response]

The following is the authors’ response to the previous reviews

Public Reviews:
**Reviewer #1 (Public review):**
(1) Authors' experimental designs have some caveats to definitely support their claims. Authors claimed that aged LT-HSCs have no myeloid-biased clone expansion using transplantation assays. In these experiments, authors used 10 HSCs and young mice as recipients. Given the huge expansion of old HSC by number and known heterogeneity in immunophenotypically defined HSC populations, it is questionable how 10 out of so many old HSCs (an average of 300,000 up to 500,000 cells per mouse; Mitchell et al., Nature Cell Biology, 2023) can faithfully represent old HSC population. The Hoxb5+ old HSC primary and secondary recipient mice data (Fig. 2C and D) support this concern. In addition, they only used young recipients. Considering the importance of inflammatory aged niche in the myeloid-biased lineage output, transplanting young vs old LT-HSCs into aged mice will complete the whole picture.

We sincerely appreciate your insightful comment regarding the existence of approximately 500,000 HSCs per mouse in older mice. To address this, we have conducted a statistical analysis to determine the appropriate sample size needed to estimate the characteristics of a population of 500,000 cells with a 95% confidence level and a ±5% margin of error. This calculation was performed using the finite population correction applied to Cochran’s formula.

For our calculations, we used a proportion of 50% (p = 0.5), as it has been reported that approximately 50% of HSCs are myeloid-biased1,2. The formula used is as follows:\begin{document}$$\displaystyle n=\frac{N \cdot Z^{2} \cdot p \cdot(1-p)}{(N-1) \cdot e^{2}+Z^{2} \cdot p \cdot(1-p)}$$\end{document}

*N* = 500,000 (total population size)

*Z* = 1.96 (*Z*-score for a 95% confidence level)

*p* = 0.5 (expected proportion)

*e* = 0.05 (margin of error)

Applying this formula, we determined that the required sample size is approximately 384 cells. This sample size ensures that the observed proportion in the sample will reflect the characteristics of the entire population. In our study, we have conducted functional experiments across Figures 2, 3, 5, 6, S3, and S6, with a total sample size of n = 126, which corresponds to over 1260 cells. While it would be ideal to analyze all 500,000 cells, this would necessitate the use of 50,000 recipient mice, which is not feasible. We believe that the number of cells analyzed is reasonable from a statistical standpoint.

References

(1) Dykstra, Brad et al. “Clonal analysis reveals multiple functional defects of aged murine hematopoietic stem cells.” The Journal of experimental medicine vol. 208,13 (2011): 2691-703. doi:10.1084/jem.20111490

(2) Beerman, Isabel et al. “Functionally distinct hematopoietic stem cells modulate hematopoietic lineage potential during aging by a mechanism of clonal expansion.” Proceedings of the National Academy of Sciences of the United States of America vol. 107,12 (2010): 5465-70. doi:10.1073/pnas.1000834107

(2) Authors' molecular data analyses need more rigor with unbiased approaches. They claimed that neither aged LT-HSCs nor aged ST-HSCs exhibited myeloid or lymphoid gene set enrichment but aged bulk HSCs, which are just a sum of LTHSCs and ST-HSCs by their gating scheme (Fig. 4A), showed the "tendency" of enrichment of myeloid-related genes based on the selected gene set (Fig. 4D). Although the proportion of ST-HSCs is reduced in bulk HSCs upon aging, since STHSCs do not exhibit lymphoid gene set enrichment based on their data, it is hard to understand how aged bulk HSCs have more myeloid gene set enrichment compared to young bulk HSCs. This bulk HSC data rather suggest that there could be a trend toward certain lineage bias (although not significant) in aged LT-HSCs or ST-HSCs. Authors need to verify the molecular lineage priming of LT-HSCs and ST-HSCs using another comprehensive dataset.

Thank you for your thoughtful feedback regarding the lack of myeloid or lymphoid gene set enrichment in aged LT-HSCs and aged ST-HSCs, despite the observed tendency for myeloid-related gene enrichment in aged bulk HSCs.

First, we acknowledge that the GSEA results vary among the different myeloid gene sets analyzed (Fig. 4, D–F; Fig. S4, C–D). Additionally, a comprehensive analysis of mouse HSC aging using multiple RNA-seq datasets reported that nearly 80% of differentially expressed genes show poor reproducibility across datasets[1]. These factors highlight the challenges of interpreting lineage bias in HSCs based solely on previously published transcriptomic data.

Given these points, we believe that emphasizing functional experimental results is more critical than incorporating an additional dataset to support our claim. In this regard, we have confirmed that young and aged LT-HSCs have similar differentiation capacity (Figure 3), while myeloid-biased hematopoiesis is observed in aged bulk HSCs (Figure S3). These findings are further corroborated by independent functional experiments. We sincerely appreciate your insightful comments.

Reference

(1) Flohr Svendsen, Arthur et al. “A comprehensive transcriptome signature of murine hematopoietic stem cell aging.” *Blood* vol. 138,6 (2021): 439-451. doi:10.1182/blood.2020009729

(3) Although authors could not find any molecular evidence for myeloid-biased hematopoiesis from old HSCs (either LT or ST), they argued that the ratio between LT-HSC and ST-HSC causes myeloid-biased hematopoiesis upon aging based on young HSC experiments (Fig. 6). However, old ST-HSC functional data showed that they barely contribute to blood production unlike young Hoxb5- HSCs (ST-HSC) in the transplantation setting (Fig. 2). Is there any evidence that in unperturbed native old hematopoiesis, old Hoxb5- HSCs (ST-HSC) still contribute to blood production?If so, what are their lineage potential/output? Without this information, it is hard to argue that the different ratio causes myeloid-biased hematopoiesis in aging context.

Thank you for the insightful and important question. The post-transplant chimerism of ST-HSCs was low in Fig. 2, indicating that transplantation induced a short-term loss of hematopoietic potential due to hematopoietic stress per cell.

To reduce this stress, we increased the number of HSCs in transplantation setting. In Fig. S6, old LT-HSCs and old ST-HSCs were transplanted in a 50:50 or 20:80 ratio, respectively. As shown in Fig. S6.D, the 20:80 group, which had a higher proportion of old ST-HSCs, exhibited a statistically significant increase in the lymphoid percentage in the peripheral blood post-transplantation.

These findings suggest that old ST-HSCs contribute to blood production following transplantation.

**Reviewer #2 (Public review):**
While aspects of their work are fascinating and might have merit, several issues weaken the overall strength of the arguments and interpretation. Multiple experiments were done with a very low number of recipient mice, showed very large standard deviations, and had no statistically detectable difference between experimental groups. While the authors conclude that these experimental groups are not different, the displayed results seem too variable to conclude anything with certainty. The sensitivity of the performed experiments (e.g. Fig 3; Fig 6C, D) is too low to detect even reasonably strong differences between experimental groups and is thus inadequate to support the author's claims. This weakness of the study is not acknowledged in the text and is also not discussed. To support their conclusions the authors need to provide higher n-numbers and provide a detailed power analysis of the transplants in the methods section.

Response #2-1:

Thank you for your important remarks. The power analysis for this experiment shows that power = 0.319, suggesting that more number may be needed. On the other hand, our method for determining the sample size in Figure 3 is as follows:

(1) First, we checked whether myeloid biased change is detected in the bulk-HSC fraction (Figure S3). The results showed that the difference in myeloid output at 16 weeks after transplantation was statistically significant (young vs. aged = 7.2 ± 8.9 vs. 42.1 ± 35.5%, p = 0.01), even though n = 10.

(2) Next, myeloid biased HSCs have been reported to be a fraction with high selfrenewal ability (2004, Blood). If myeloid biased HSCs increase with aging, the increase in myeloid biased HSCs in LT-HSC fraction would be detected with higher sensitivity than in the bulk-HSC fraction used in Figure S3.

(3) However, there was no difference not only in p-values but also in the mean itself, young vs aged = 51.4±31.5% vs 47.4±39.0%, p = 0.82, even though n = 8 in Figure 3. Since there was no difference in the mean itself, it is highly likely that no difference will be detected even if n is further increased.

Regarding Figure 6, we obtained a statistically significant difference and consider the sample size to be sufficient. In addition, we have performed various functional experiments (Figures 2, 5, 6 and S6), and have obtained consistent results that expansion of myeloid biased HSCs does not occur with aging in Hoxb5+HSCs fraction. Based on the above, we conclude that the LT-HSC fraction does not differ in myeloid differentiation potential with aging.

As the authors attempt to challenge the current model of the age-associated expansion of myeloid-biased HSCs (which has been observed and reproduced by many different groups), ideally additional strong evidence in the form of single-cell transplants is provided.

Response #2-2:

Thank you for the comments. As the reviewer pointed out, we hope we could reconfirm our results using single-cell level technology in the future.

On the other hand, we have reported that the ratio of myeloid to lymphoid cells in the peripheral blood changes when the number of HSCs transplanted, or the number of supporting cells transplanted with HSCs, is varied[1-2]. Therefore, single-cell transplant data need to be interpreted very carefully to determine differentiation potential.

From this viewpoint, future experiments will combine the Hoxb5 reporter system with a lineage tracing system that can track HSCs at the single-cell level over time. This approach will investigate changes in the self-renewal capacity of individual HSCs and their subsequent differentiation into progenitor cells and peripheral blood cells. We have reflected this comment by adding the following sentences in the manuscript.

[P19, L451] “In contrast, our findings should be considered in light of some limitations. In this report, we primarily performed ten to twenty cell transplantation assays. Therefore, the current theory should be revalidated using single-cell technology with lineage tracing system[3-4]. This approach will investigate changes in the self-renewal capacity of individual HSCs and their subsequent differentiation into progenitor cells and peripheral blood cells.”

It is also unclear why the authors believe that the observed reduction of ST-HSCs relative to LT-HSCs explains the myeloid-biased phenotype observed in the peripheral blood. This point seems counterintuitive and requires further explanation.

Response #2-3:

Thank you for your comment. We apologize for the insufficient explanation. Our data, as shown in Figures 3 and 4, demonstrate that the differentiation potential of LT-HSCs remains unchanged with age. Therefore, rather than suggesting that an increase in LT-HSCs with a consistent differentiation capacity leads to myeloidbiased hematopoiesis, it seems more accurate to highlight that the relative decrease in the proportion of ST-HSCs, which remain in peripheral blood as lymphocytes, leads to a relative increase in myeloid cells in peripheral blood and thus causes myeloid-biased hematopoiesis.

However, if we focus on the increase in the ratio of LT-HSCs, it is also plausible to explain that “with aging, the proportion of LT-HSCs capable of long-term myeloid hematopoiesis increases. As a result, from 16 weeks after transplantation, the influence of LT-HSCs maintaining the long-term ability to produce myeloid cells becomes relatively more significant, leading to an increase in the ratio of myeloid cells in the peripheral blood and causing myeloid-biased hematopoiesis.”

Based on my understanding of the presented data, the authors argue that myeloidbiased HSCs do not exist, asa) they detect no difference between young/aged HSCs after transplant (mind low nnumbers and large std!!!); b) myeloid progenitors downstream of HSCs only show minor or no changes in frequency and c) aged LT-HSCs do not outperform young LT-HSC in myeloid output LT-HSCs in competitive transplants (mind low n-numbers and large std!!!).However, given the low n-numbers and high variance of the results, the argument seems weak and the presented data does not support the claims sufficiently. That the number of downstream progenitors does not change could be explained by other mechanisms, for instance, the frequently reported differentiation short-cuts of HSCs and/or changes in the microenvironment.

Response #2-4:

We appreciate the comments. As mentioned above, we will correct the manuscript regarding the sample size. Regarding the interpreting of the lack of increase in the percentage of myeloid progenitor cells in the bone marrow with age, it is instead possible that various confounding factors, such as differentiation shortcuts or changes in the microenvironment, are involved.

However, even when aged LT-HSCs and young LT-HSCs are transplanted into the same recipient mice, the timing of the appearance of different cell fractions in peripheral blood is similar (Figure 3 of this paper). Therefore, we have not obtained data suggesting that clear shortcuts exist in the differentiation process of aged HSCs into neutrophils or monocytes. Additionally, it is currently consensually accepted that myeloid cells, including neutrophils and monocytes, differentiate from GMPs[1]. Since there is no changes in the proportion of GMPs in the bone marrow with age, we concluded that the differentiation potential into myeloid cells remains consistent with aging.

"Then, we found that the myeloid lineage proportions from young and aged LT-HSCs were nearly comparable during the observation period after transplantation (Fig. 3, B and C)."

[Comment to the authors]: Given the large standard deviation and low n-numbers, the power of the analysis to detect differences between experimental groups is very low. Experimental groups with too large standard deviations (as displayed here) are difficult to interpret and might be inconclusive. The absence of clearly detectable differences between young and aged transplanted HSCs could thus simply be a false-negative result. The shown experimental results hence do not provide strong evidence for the author's interpretation of the data. The authors should add additional transplants and include a detailed power analysis to be able to detect differences between experimental groups with reasonable sensitivity.

Response #2-5:

Thank you for providing these insights. Regarding the sample size, we have addressed this in Response #2-1.

Line 293: "Based on these findings, we concluded that myeloid-biased hematopoiesis observed following transplantation of aged HSCs was caused by a relative decrease in ST-HSC in the bulk-HSC compartment in aged mice rather than the selective expansion of myeloid-biased HSC clones."Couldn't that also be explained by an increase in myeloid-biased HSCs, as repeatedly reported and seen in the expansion of CD150+ HSCs? It is not intuitively clear why a reduction of ST-HSCs clones would lead to a myeloid bias. The author should try to explain more clearly where they believe the increased number of myeloid cells comes from. What is the source of myeloid cells if the authors believe they are not derived from the expanded population of myeloid-biased HSCs? t

Response #2-6:

Thank you for pointing this out. We apologize for the insufficient explanation. We will explain using Figure 8 from the paper.

First, our data show that LT-HSCs maintain their differentiation capacity with age, while ST-HSCs lose their self-renewal capacity earlier, so that only long-lived memory lymphocytes remain in the peripheral blood after the loss of selfrenewal capacity in ST-HSCs (Figure 8, upper panel). In mouse bone marrow, the proportion of LT-HSCs increases with age, while the proportion of ST-HSCs relatively decreases (Figure 8, lower panel and Figure S5).

Our data show that merely reproducing the ratio of LT-HSCs to ST-HSCs observed in aged mice using young LT-HSCs and ST-HSCs can replicate myeloidbiased hematopoiesis. This suggests that the increase in LT-HSC and the relative decrease in ST-HSC within the HSC compartment with aging are likely to contribute to myeloid-biased hematopoiesis.

As mentioned earlier, since the differentiation capacity of LT-HSCs remain unchaged with age, it seems more accurate to describe that the relative decrease in the proportion of ST-HSCs, which retain long-lived memory lymphocytes in peripheral blood, leads to a relative increase in myeloid cells in peripheral blood and thus causes myeloid-biased hematopoiesis.

However, focusing on the increase in the proportion of LT-HSCs, it is also possible to explain that “with aging, the proportion of LT-HSCs capable of long-term myeloid hematopoiesis increases. As a result, from 16 weeks after transplantation, the influence of LT-HSCs maintaining the long-term ability to produce myeloid cells becomes relatively more significant, leading to an increase in the ratio of myeloid cells in the peripheral blood and causing myeloid-biased hematopoiesis.”

**Recommendations for the authors:**

**Reviewer #2 (Recommendations for the authors):**
Summary:Comment #2-1: While aspects of their work are fascinating and might have merit, several issues weaken the overall strength of the arguments and interpretation. Multiple experiments were done with a very low number of recipient mice, showed very large standard deviations, and had no statistically detectable difference between experimental groups. While the authors conclude that these experimental groups are not different, the displayed results seem too variable to conclude anything with certainty. The sensitivity of the performed experiments (e.g. Figure 3; Figure 6C, D) is too low to detect even reasonably strong differences between experimental groups and is thus inadequate to support the author's claims. This weakness of the study is not acknowledged in the text and is also not discussed. To support their conclusions the authors, need to provide higher n-numbers and provide a detailed power analysis of the transplants in the methods section.

Response #2-1

Thank you for your important remarks. The power analysis for this experiment shows that power = 0.319, suggesting that more number may be needed. On the other hand, our method for determining the sample size in Figure 3 is as follows:

(1) First, we checked whether myeloid biased change is detected in the bulk-HSC fraction (Figure S3). The results showed that the difference in myeloid output at 16 weeks after transplantation was statistically significant (young vs. aged = 7.2 {plus minus} 8.9 vs. 42.1 {plus minus} 35.5%, p = 0.01), even though n = 10.

(2) Next, myeloid biased HSCs have been reported to be a fraction with high selfrenewal ability (2004, Blood). If myeloid biased HSCs increase with aging, the increase in myeloid biased HSCs in LT-HSC fraction would be detected with higher sensitivity than in the bulk-HSC fraction used in Figure S3.

(3) However, there was no difference not only in p-values but also in the mean itself, young vs aged = 51.4{plus minus}31.5% vs 47.4{plus minus}39.0%, p = 0.82, even though n = 8 in Figure 3. Since there was no difference in the mean itself, it is highly likely that no difference will be detected even if n is further increased.

Regarding Figure 6, we obtained a statistically significant difference and consider the sample size to be sufficient. In addition, we have performed various functional experiments (Figures 2, 5, 6 and S6), and have obtained consistent results that expansion of myeloid-biased HSCs does not occur with aging in Hoxb5+HSCs fraction. Based on the above, we conclude that the LT-HSC fraction does not differ in myeloid differentiation potential with aging.

[Comment for authors]Paradigm-shifting extraordinary claims require extraordinary data. Unfortunately, the authors do not provide additional data to further support their claims. Instead, the authors argue the following: Because they were able to find significant differences between experimental groups in some experiments, the absence of significant differences in the results of other experiments must be correct, too.

This logic is in my view flawed. Any assay/experiment with highly variable data has a very low sensitivity to detect significant differences between groups. If, as in this case, the variance is as large as the entire dynamic range of the readout, it becomes impossible to be able to detect any difference. In these cases, it is not surprising and actually expected that the mean of the group is located close to the center of the dynamic range as is the case here (center of dynamic range: 50%). In other words, this means that the experiments are simply not reproducible. It is absolutely critical to remember that any experiment and its associated statistical analysis has 3 (!!!) instead of 2 possible outcomes:

(1) There is a statistically significant difference

(2) There is no statistically significant difference

(3) The results of the experiment are inconclusive because the replicates are too variable and the results are not reproducible.

While most of us are inclined to think about outcomes (1) or (2), outcome (3) cannot be neglected. While it might be painful to accept, the only way to address concerns about data reproducibility is to provide additional data, improve reproducibility, and lower the power of the analysis to an acceptable level (e.g. able to detect difference of 5-10% between groups).

Without going into the technical details, the example graph from the link below illustrates that with a power 0.319 as stated by the authors, approx. 25 transplants, instead of 8, would be required.

Typically, however, a power of 0.8 is a reasonable value for any power analysis (although it's not a very strong power either). Even if we are optimistic and assume that there might be a reasonably large difference between experimental groups (in the example above P2 = 0.6, which is actually not that large) we can estimate that we would need over 10 transplants per group to say with confidence that two experimental groups likely do not differ. With smaller differences, these numbers increase quickly to 20+ transplants per group as can be seen in the example graph using an Alpha of 0.1 above.

Further reading can be found here and in many textbooks or other online resources: https://power-analysis.com/effect_size.htm%20https://tss.awf.poznan.pl/pdf-188978-110207?filename=Using%20power%20analysis%20to.pdf.

Response:

Thank you for your feedback. We fully agree with the reviewer that paradigmshifting claims must be supported by equally robust data. It has been welldocumented that the frequency of myeloid-biased HSCs increases with age, with reports indicating that over 50% of the HSC compartment in aged mice consists of myeloid-biased HSCs[1,2]. Based on this, we believe that if aged LT-HSCs were substantially myeloid-biased, the difference should be readily detectable.

To further validate our findings, we showed the similar preliminary experiment. The resulting data are shown below (n = 8).

**Author response image 1. sa4fig1:** (A) Experimental design for competitive co-transplantation assay. Ten CD45.2^+^ young LT-HSCs and ten CD45.2^+^ aged LT-HSCs were transplanted with 2 × 10^5^ CD45.1^+^/CD45.2^+^ supporting cells into lethally irradiated CD45.1^+^ recipient mice (*n* = 8). (B) Lineage output of young or aged LT-HSCs at 4, 8, 12, 16 weeks after transplantation. Each bar represents an individual mouse. **P* < 0.05. ***P* < 0.01.

While a slight increase in myeloid-biased hematopoiesis was observed in the aged LT-HSC fraction, the difference was not statistically significant. These new results are presented alongside the original Figure 3, which was generated using a larger sample size (n = 16).

**Author response image 2. sa4fig2:** (A) Experimental design for competitive co-transplantation assay.Ten CD45.2^+^ young LT-HSCs and ten CD45.2^+^ aged LT-HSCs were transplanted with 2 × 10^5^ CD45.1^+^/CD45.2^+^ supporting cells into lethally irradiated CD45.1^+^ recipient mice (*n* = 16). (B) Lineage output of young or aged LT-HSCs at 4, 8, 12, 16 weeks after transplantation. Each bar represents an individual mouse.

Consistent with the original data, aged LT-HSCs exhibited a lineage output that was nearly identical to that of young LT-HSCs. Nonetheless, as the reviewer rightly pointed out, we cannot completely exclude the possibility that subtle differences may exist but remain undetected. To address this, we have added the following sentence to the manuscript:

[P9, L200] “These findings unmistakably demonstrated that mixed/bulk-HSCs showed myeloid skewed hematopoiesis in PB with aging. In contrast, LT-HSCs maintained a consistent lineage output throughout life, although subtle differences between aged and young LT-HSCs may exist and cannot be entirely ruled out.”

References

(1) Dykstra, Brad et al. “Clonal analysis reveals multiple functional defects of aged murine hematopoietic stem cells.” The Journal of experimental medicine vol. 208,13 (2011): 2691-703. doi:10.1084/jem.20111490

(2) Beerman, Isabel et al. “Functionally distinct hematopoietic stem cells modulate hematopoietic lineage potential during aging by a mechanism of clonal expansion.” Proceedings of the National Academy of Sciences of the United States of America vol. 107,12 (2010): 5465-70. doi:10.1073/pnas.1000834107

Comment #2-3: It is also unclear why the authors believe that the observed reduction of STHSCs relative to LT-HSCs explains the myeloid-biased phenotype observed in the peripheral blood. This point seems counterintuitive and requires further explanation.

Response #2-3:

Thank you for your comment. We apologize for the insufficient explanation. Our data, as shown in Figures 3 and 4, demonstrate that the differentiation potential of LTHSCs remains unchanged with age. Therefore, rather than suggesting that an increase in LT-HSCs with a consistent differentiation capacity leads to myeloid biased hematopoiesis, it seems more accurate to highlight that the relative decrease in the proportion of ST-HSCs, which remain in peripheral blood as lymphocytes, leads to a relative increase in myeloid cells in peripheral blood and thus causes myeloid-biased hematopoiesis. However, if we focus on the increase in the ratio of LT-HSCs, it is also plausible to explain that "with aging, the proportion of LT-HSCs capable of long-term myeloid hematopoiesis increases. As a result, from 16 weeks after transplantation, the influence of LT-HSCs maintaining the long-term ability to produce myeloid cells becomes relatively more significant, leading to an increase in the ratio of myeloid cells in the peripheral blood and causing myeloid-biased hematopoiesis."

[Comment for authors]While this interpretation of the data might make sense the shown data do not exclude alternative explanations. The authors do not exclude the possibility that LTHSCs expand with age and that this expansion in combination with an aging microenvironment drives myeloid bias. The authors should quantify the frequency [%] and absolute number of LT-HSCs and ST-HSCs in young vs. aged animals. Especially analyzing the abs. numbers of cells will be important to support their claims as % can be affected by changes in the frequency of other populations.

Thank you for your very important point. As this reviewer pointed out, we do not exclude the possibility that the combination of aged microenvironment drives myeloid bias. Additionally, we acknowledge that myeloid-biased hematopoiesis with age is a complex process likely influenced by multiple factors. We would like to discuss the mechanism mentioned as a future research direction. Thank you for the insightful feedback. Regarding the point about the absolute cell numbers mentioned in the latter half of the paragraph, we will address this in detail in our subsequent response (Response #2-4).

Comment #2-4: Based on my understanding of the presented data, the authors argue that myeloid-biased HSCs do not exist, as (a) they detect no difference between young/aged HSCs after transplant (mind low n-numbers and large std!); (b) myeloid progenitors downstream of HSCs only show minor or no changes in frequency and (c) aged LT-HSCs do not outperform young LT-HSCs in myeloid output LTHSCs in competitive transplants (mind low n-numbers and large std!). However, given the low n-numbers and high variance of the results, the argument seems weak and the presented data does not support the claims sufficiently. That the number of downstream progenitors does not change could be explained by other mechanisms, for instance, the frequently reported differentiation short-cuts of HSCs and/or changes in the microenvironment.

Response #2-4:

We appreciate the comments. As mentioned above, we will correct the manuscript regarding the sample size. Regarding the interpreting of the lack of increase in the percentage of myeloid progenitor cells in the bone marrow with age, it is instead possible that various confounding factors, such as differentiation shortcuts or changes in the microenviroment, are involved. However, even when aged LT-HSCs and young LT-HSCs are transplanted into the same recipient mice, the timing of the appearance of different cell fractions in peripheral blood is similar (Figure 3 of this paper). Therefore, we have not obtained data suggesting that clear shortcuts exist in the differentiation process of aged HSCs into neutrophils or monocytes. Additionally, it is currently consensually accepted that myeloid cells, including neutrophils and monocytes, differentiate from GMPs1. Since there are no changes in the proportion of GMPs in the bone marrow with age, we concluded that the differentiation potential into myeloid cells remains consistent with aging.

Reference

(1) Akashi K and others, 'A Clonogenic Common Myeloid Progenitor That Gives Rise to All Myeloid Lineages', Nature, 404.6774 (2000), 193-97.

[Comment for authors]As the relative frequency of cell population can be misleading, the authors should compare the absolute numbers of progenitors in young vs. aged mice to strengthen their argument. It would also be helpful to quantify the absolute numbers and relative frequencies in WT mice to exclude the possibility the HoxB5-trimcherry mouse model suffers from unexpected aging phenotypes and the hematopoietic system differs from wild-type animals.

Thank you for your valuable feedback. We understand the importance of comparing the absolute numbers of progenitors in young versus aged mice to provide a more accurate representation of the changes in cell populations.

Therefore, we quantified the absolute cell count of hematopoietic cells in the bone marrow using flow cytometry data.

**Author response image 3. sa4fig3:** 

As previously reported, we observed a 10-fold increase in the number of pHSCs in aged mice compared to young mice. Additionally, our analysis revealed a statistically significant decrease in the number of Flk2+ progenitors and CLPs in aged mice. On the other hand, there was no statistically significant change in the number of myeloid progenitors between the two age groups. We appreciate the suggestion and hope that this additional information strengthens our argument and addresses your concerns.

Comment #2-5:"Then, we found that the myeloid lineage proportions from young and aged LT-HSCs were nearly comparable during the observation period after transplantation (Figure 3, B and C)." Given the large standard deviation and low n-numbers, the power of the analysis to detect differences between experimental groups is very low. Experimental groups with too large standard deviations (as displayed here) are difficult to interpret and might be inconclusive. The absence of clearly detectable differences between young and aged transplanted HSCs could thus simply be a false-negative result. The shown experimental results hence do not provide strong evidence for the author's interpretation of the data. The authors should add additional transplants and include a detailed power analysis to be able to detect differences between experimental groups with reasonable sensitivity.

Response #2-5:

Thank you for providing these insights. Regarding the sample size, we have addressed this in Response #2-1.

[Comment for authors]As explained in detail in the response to #2-1 the provided arguments are not convincing. As the authors pointed out, the power of these experiments is too low to make strong claims. If the author does not intend to provide new data, the language of the manuscript needs to be adjusted to reflect this weakness. A paragraph discussing the limitations of the study mentioning the limited power of the data should be included beyond the above-mentioned rather vague statement that the data should be validated (which is almost always necessary anyway).

Thank you for your valuable comment. We agree with the importance of discussing potential limitations in our experimental design. In response to the reviewer’s suggestion, we have revised the manuscript to include the following sentences:

[P19, L434] "In the co-transplantation assay shown in Figure 3, the myeloid lineage output derived from young and aged LT-HSCs was comparable (Young LT-HSC: 51.4 ± 31.5% vs. Aged LT-HSC: 47.4 ± 39.0%, *p* = 0.82). Although no significant difference was detected, the small sample size (*n* = 8) may limit the sensitivity of the assay to detect subtle myeloid-biased phenotypes."

This addition acknowledges the potential limitations of our analysis and highlights the need for further investigation with larger cohorts.

Comment #2-6:Line 293: "Based on these findings, we concluded that myeloid biased hematopoiesis observed following transplantation of aged HSCs was caused by a relative decrease in ST-HSC in the bulk-HSC compartment in aged mice rather than the selective expansion of myeloid-biased HSC clones." Couldn't that also be explained by an increase in myeloid-biased HSCs, as repeatedly reported and seen in the expansion of CD150+ HSCs? It is not intuitively clear why a reduction of STHSCs clones would lead to a myeloid bias. The author should try to explain more clearly where they believe the increased number of myeloid cells comes from. What is the source of myeloid cells if the authors believe they are not derived from the expanded population of myeloid-biased HSCs?

Response #2-6:

Thank you for pointing this out. We apologize for the insufficient explanation. We will explain using attached Figure 8 from the paper. First, our data show that LT-HSCs maintain their differentiation capacity with age, while ST-HSCs lose their self-renewal capacity earlier, so that only long-lived memory lymphocytes remain in the peripheral blood after the loss of self-renewal capacity in ST-HSCs (Figure 8, upper panel). In mouse bone marrow, the proportion of LT-HSCs increases with age, while the proportion of STHSCs relatively decreases (Figure 8, lower panel and Figure S5).

Our data show that merely reproducing the ratio of LT-HSCs to ST-HSCs observed in aged mice using young LT-HSCs and ST-HSCs can replicate myeloid-biased hematopoiesis. This suggests that the increase in LT-HSC and the relative decrease in ST-HSC within the HSC compartment with aging are likely to contribute to myeloid-biased hematopoiesis.

As mentioned earlier, since the differentiation capacity of LT-HSCs remain unchanged with age, it seems more accurate to describe that the relative decrease in the proportion of STHSCs, which retain long-lived memory lymphocytes in peripheral blood, leading to a relative increase in myeloid cells in peripheral blood and thus causes myeloid-biased hematopoiesis. However, focusing on the increase in the proportion of LT-HSCs, it is also possible to explain that "with aging, the proportion of LT-HSCs capable of long-term myeloid hematopoiesis increases. As a result, from 16 weeks after transplantation, the influence of LT-HSCs maintaining the long-term ability to produce myeloid cells become relatively more significant, leading to an increase in the ratio of myeloid cells in the peripheral blood and causing myeloid biased hematopoiesis."

[Comment for authors]While I can follow the logic of the argument, my concerns about the interpretation remain as I see discrepancies in other findings in the published literature. For instance, what the authors call ST-HSCs, differs from the classical functional definition of ST-HSCs. It is thus difficult to relate the described observations to previous reports. ST-HSCs typically can contribute significantly to multiple lineages for several weeks (see for example PMID: 29625072). It is somewhat surprising that the ST-HSC in this study don't show this potential and loose their potential much quicker.The authors should thus provide a more comprehensive depth of immunophenotypic and molecular characterization to compare their LT-HSCs to ST-HSCs. For instance, are LT-HSCs CD41- HSCs? How do ST-HSCs differ in their surface marker expression from previously used definitions of ST-HSCs? A list of differentially expressed genes between young and old LT-HSCs and ST-HSCs should be done and will likely provide important insights into the molecular programs/markers (beyond the provided GO analysis, which seems superficial).

Thank you for your valuable feedback. As the reviewer noted, there are indeed multiple definitions of ST-HSCs. We appreciate the opportunity to clarify our definitions of ST-HSCs. We define ST-HSCs functionally, rather than by surface antigens, which we believe is the most classical and widely accepted definition [1]. In our study, we define long-term hematopoietic stem cells (LT-HSCs) as those HSCs that continue to contribute to hematopoiesis after a second transplantation and possess long-term self-renewal potential. Conversely, we define short-term hematopoietic stem cells (ST-HSCs) as those HSCs that do not contribute to hematopoiesis after a second transplantation and only exhibit self-renewal potential in the short term.

Next, in the paper referenced by the reviewer[2], the chimerism of each fraction of ST-HSCs also peaked at 4 weeks and then decreased to approximately 0.1% after 12 weeks post-transplantation. Author response image 5 illustrates our ST-HSC donor chimerism in Figure 2. We believe that data in the paper referenced by the reviewer2 is consistent with our own observations of the hematopoietic pattern following ST-HSC transplantation, indicating a characteristic loss of hematopoietic potential 4 weeks after the transplantation. Furthermore, as shown in Figures 2D and 2F, the fraction of ST-HSCs does not exhibit hematopoietic activity after the second transplantation. Therefore, we consider this fraction to be ST-HSCs.

**Author response image 4. sa4fig4:** 

Additionally, the RNAseq data presented in Figures 4 and S4 revealed that the GSEA results vary among the different myeloid gene sets analyzed (Fig. 4, D–F; Fig. S4, C–D). Moreover, a comprehensive analysis of mouse HSC aging using multiple RNA-seq datasets reported that nearly 80% of differentially expressed genes show poor reproducibility across datasets[3]. From the above, while RNAseq data is indeed helpful, we believe that emphasizing functional experimental results is more critical than incorporating an additional dataset to support our claim. Thank you once again for your insightful feedback.

References

(1) Kiel, Mark J et al. “SLAM family receptors distinguish hematopoietic stem and progenitor cells and reveal endothelial niches for stem cells.” Cell vol. 121,7 (2005): 1109-21. doi:10.1016/j.cell.2005.05.026

(2) Yamamoto, Ryo et al. “Large-Scale Clonal Analysis Resolves Aging of the Mouse Hematopoietic Stem Cell Compartment.” Cell stem cell vol. 22,4 (2018): 600-607.e4. doi:10.1016/j.stem.2018.03.013

(3) Flohr Svendsen, Arthur et al. “A comprehensive transcriptome signature of murine hematopoietic stem cell aging.” Blood vol. 138,6 (2021): 439-451. doi:10.1182/blood.2020009729

**Reviewer #3 (Public review):**
Although the topic is appropriate and the new model provides a new way to think about lineage-biased output observed in multiple hematopoietic contexts, some of the experimental design choices, as well as some of the conclusions drawn from the results could be substantially improved. Also, they do not propose any potential mechanism to explain this process, which reduces the potential impact and novelty of the study.

The authors have satisfactorily replied to some of my comments. However, there are multiple key aspects that still remain unresolved.

**Reviewer #3 (Recommendations for the authors):**
Comment #3-1,2:Although the additional details are much appreciated the core of my original comments remains unanswered. There are still no details about the irradiation dose for each particular experiment. Is any transplant performed using a 9.1 Gy dose? If yes, please indicate it in text or figure legend. If not, please remove this number from the corresponding method section.Again, 9.5 Gy (split in two doses) is commonly reported as sublethal. The fact that the authors used a methodology that deviates from the "standard" for the field makes difficult to put these results in context with previous studies. It is not possible to know if the direct and indirect effects of this conditioning method in the hematopoietic system have any consequences in the presented results.

Thank you for your clarification. We confirm that none of the transplantation experiments described were performed using a 9.1 Gy irradiation dose. We have therefore removed the mention of "9.1 Gy" from the relevant section of the Materials and Methods. We appreciate helpful suggestion to improve the clarity of the manuscript.

[P22, L493] “12-24 hours prior to transplantation, C57BL/6-Ly5.1 mice, or aged C57BL/6J recipient mice were lethally irradiated with single doses of 8.7 Gy.”

Regarding the reviewer’s concern about the radiation dose used in our experiments, we will address this point in more detail in our subsequent response (see Response #3-4).

Comment #3-4(Original): When representing the contribution to PB from transplanted cells, the authors show the % of each lineage within the donor-derived cells (Figures 3B-C, 5B, 6B-D, 7C-E, and S3 B-C). To have a better picture of total donor contribution, total PB and BM chimerism should be included for each transplantation assay. Also, for Figures 2C-D and Figures S2A-B, do the graphs represent 100% of the PB cells? Are there any radioresistant cells?Response #3-4 (Original): Thank you for highlighting this point. Indeed, donor contribution to total peripheral blood (PB) is important information. We have included the donor contribution data for each figure above mentioned.In Figure 2C-D and Figure S2A-B, the percentage of donor chimerism in PB was defined as the percentage of CD45.1-CD45.2+ cells among total CD45.1-CD45.2+ and CD45.1+CD45.2+ cells as described in method section.

Comment for our #3-4 response:

Thanks for sharing these data. These graphs should be included in their corresponding figures along with donor contribution to BM.

Regarding Figure2 C-D, as currently shown, the graphs only account for CD45.1CD45.2+ (donor-derived) and CD45.1+CD45.2+ (supporting-derived). What is the percentage of CD45.1+CD45.2- (recipient-derived)? Since the irradiation regiment is atypical, including this information would help to know more about the effects of this conditioning method.

Thank you for your insightful comment regarding Figure 2C-D. To address the concern that the reviewer pointed out, we provide the kinetics of the percentage of CD45.1+CD45.2- (recipient-derived) in Author response image 7.

**Author response image 5. sa4fig5:** 

As the reviewer pointed out, we observed the persistence of recipient-derived cells, particularly in the secondary transplant. As noted, this suggests that our conditioning regimen may have been suboptimal. In response, we will include the donor chimerism analysis in the total cells and add the following statement in the study limitations section to acknowledge this point:

[P19, L439] “Additionally, in this study, we purified LT-HSCs using the Hoxb5 reporter system and employed a moderate conditioning regimen (8.7 Gy). To have a better picture of total donor contribution, total PB chimerism are presented in Figure S7 and we cannot exclude the possibility that these factors may have influenced the results. Therefore, it would be ideal to validate our findings using alternative LT-HSC markers and different conditioning regimens.”

Comment #3-5: For BM progenitor frequencies, the authors present the data as the frequency of cKit+ cells. This normalization might be misleading as changes in the proportion of cKit+ between the different experimental conditions could mask differences in these BM subpopulations. Representing this data as the frequency of BM single cells or as absolute numbers (e.g., per femur) would be valuable.

Response #3-5:

We appreciate the reviewer's comment on this point.

Firstly, as shown in Supplemental Figures S1B and S1C, we analyze the upstream (HSC, MPP, Flk2+) and downstream (CLP, MEP, CMP, GMP) fractions in different panels. Therefore, normalization is required to assess the differentiation of HSCs from upstream to downstream.

Additionally, the reason for normalizing by c-Kit+ is that the bone marrow analysis was performed after enrichment using the Anti-c-Kit antibody for both upstream and downstream fractions. Based on this, we calculated the progenitor populations as a frequency within the c-Kit positive cells. Next, the results of normalizing the whole bone marrow cells (live cells) are shown below.

**Author response image 6. sa4fig6:** 

Similar to the results of normalizing c-Kit+ cells, myeloid progenitors remained unchanged, including a statistically significant decrease in CMP in aged mice. Additionally, there were no significant differences in CLP. In conclusion, similar results were obtained between the normalization with c-Kit and the normalization with whole bone marrow cells (live cells).

However, as the reviewer pointed out, it is necessary to explain the reason for normalization with c-Kit. Therefore, we will add the following description.

[P21, L502] For the combined analysis of the upstream (HSC, MPP, Flk2+) and downstream (CLP, MEP, CMP, GMP) fractions in Figures 1B, we normalized by cKit+ cells because we performed a c-Kit enrichment for the bone marrow analysis.

Comment for our #3-5 response:

I understand that normalization is necessary to compare across different BM populations. However, the best way would be to normalize to single cells. As I mentioned in my original comment, normalizing to cKit+ cells could be misleading, as the proportion of cKit+ cells could be different across the experimental conditions. Further, enriching for cKit+ cells when analyzing BM subpopulation frequencies could introduce similar potential errors. The enrichment would depend on the level of expression of cKit for each of these population, what would alter the final quantification. Indeed, CLP are typically defined as cKit-med/low. Thus, cKit enrichment would not be a great method to analyze the frequency of these cells.

The graph in the authors' response to my comment, show similar trend to what is represented Figure 1B for some populations. However, there are multiple statistically significant changes that disappear in this new version. This supports my original concern and, in consequence, I would encourage to represent this data as the frequency of BM single cells or as absolute numbers (e.g., per femur).

Thank you for your thoughtful follow-up comment. In response to the reviewer’s suggestion, we will represent the data as the frequency among total BM single cells. These revised graphs have been incorporated into the updated Figure 7F and corresponding figure legend have been revised accordingly to accurately reflect these representations. We appreciate your valuable input, which has helped us improve the clarity and rigor of our data presentation.

Comment #3-6: Regarding Figure 1B, the authors argue that if myeloid-biased HSC clones increase with age, they should see increased frequency of all components of the myeloid differentiation pathway (CMP, GMP, MEP). This would imply that their results (no changes or reduction in these myeloid subpopulations) suggest the absence of myeloid-biased HSC clones expansion with age. This reviewer believes that differentiation dynamics within the hematopoietic hierarchy can be more complex than a cascade of sequential and compartmentalized events (e.g., accelerated differentiation at the CMP level could cause exhaustion of this compartment and explain its reduction with age and why GMP and MEP are unchanged) and these conclusions should be considered more carefully.

Response #3-6:

We wish to thank the reviewer for this comment. We agree with that the differentiation pathway may not be a cascade of sequential events but could be influenced by various factors such as extrinsic factors.

In Figure 1B, we hypothesized that there may be other mechanisms causing myeloid-biased hematopoiesis besides the age-related increase in myeloid-biased HSCs, given that the percentage of myeloid progenitor cells in the bone marrow did not change with age. However, we do not discuss the presence or absence of myeloid-biased HSCs based on the data in Figure 1B.

Our newly proposed theories—that the differentiation capacity of LT-HSCs remains unchanged with age and that age-related myeloid-biased hematopoiesis is due to changes in the ratio of LT-HSCs to ST-HSCs—are based on functional experiment results. As the reviewer pointed out, to discuss the presence or absence of myeloid-biased HSCs based on the data in Figure 1B, it is necessary to apply a system that can track HSC differentiation at single-cell level. The technology would clarify changes in the self-renewal capacity of individual HSCs and their differentiation into progenitor cells and peripheral blood cells. The authors believe that those single-cell technologies will be beneficial in understanding the differentiation of HSCs. Based on the above, the following statement has been added to the text.

[P19, L440] In contrast, our findings should be considered in light of some limitations. In this report, we primarily performed ten to twenty cell transplantation assays. Therefore, the current theory should be revalidated using single-cell technology with lineage tracing system1-2. This approach will investigate changes in the self-renewal capacity of individual HSCs and their subsequent differentiation into progenitor cells and peripheral blood cells.

Comment for our #3-6 response:

Thanks for the response. My original comments referred to the statement "On the other hand, in contrast to what we anticipated, the frequency of GMP was stable, and the percentage of CMP actually decreased significantly with age, defying our prediction that the frequency of components of the myeloid differentiation pathway, such as CMP, GMP, and MEP would increase in aged mice if myeloid-biased HSC clones increase with age (Fig. 1 B)" (lines #129-133). Again, the absence of an increase in CMP, GMP and MEP with age does not mean the absence of and increase in myeloid-biased HSC clones. This statement should be considered more carefully.

Thank you for the insightful comment. We agree that the absence of an increase in CMP, GMP and MEP with age does not mean the absence of an increase in myeloid-biased HSC clones. In our revised manuscript, we have refined the statement to acknowledge this nuance more clearly. The updated text now reads as follows:

P6, L129 On the other hand, in contrast to what we anticipated, the frequency of GMP was stable, and the percentage of CMP actually decreased significantly with age, defying our prediction that the frequency of components of the myeloid differentiation pathway, such as CMP, GMP, and MEP may increase in aged mice, if myeloid-biased HSC clones increase with age.

Comment #3-7: Within the few recipients showing good donor engraftment in Figure 2C, there is a big proportion of T cells that are "amplified" upon secondary transplantation (Figure 2D). Is this expected?

Response #3-7:

We wish to express our deep appreciation to the reviewer for insightful comment on this point. As the reviewers pointed out, in Figure 2D, a few recipients show a very high percentage of T cells. The authors had the same question and considered this phenomenon as follows:

(1) One reason for the very high percentage of T cells is that we used 1 x 107 whole bone marrow cells in the secondary transplantation. Consequently, the donor cells in the secondary transplantation contained more T-cell progenitor cells, leading to a greater increase in T cells compared to the primary transplantation.

(2) We also consider that this phenomenon may be influenced by the reduced selfrenewal capacity of aged LT-HSCs, resulting in decreased sustained production of myeloid cells in the secondary recipient mice. As a result, long-lived memorytype lymphocytes may preferentially remain in the peripheral blood, increasing the percentage of T cells in the secondary recipient mice.

We have discussed our hypothesis regarding this interesting phenomenon. To further clarify the characteristics of the increased T-cell count in the secondary recipient mice, we will analyze TCR clonality and diversity in the future.

Comment for our #3-7 response:

Thanks for the potential explanations to my question. This fact is not commonly reported in previous transplantation studies using aged HSCs. Could Hoxb5 label fraction of HSCs that is lymphoid/T-cell biased upon secondary transplantation? The number of recipients with high frequency of lymphoid cells in the peripheral blood (even from young mice) is remarkable.

Response:

Thank you for your insightful suggestion. Based on this comment, we calculated the percentage of lymphoid cells in the donor fraction at 16 weeks following the secondary transplantation, which was 56.1 ± 25.8% (L/M = 1.27). According to the Müller-Sieburg criteria, lymphoid-biased hematopoiesis is defined as having an L/M ratio greater than 10.

Given our findings, we concluded that the Hoxb5-labeled fraction does not specifically indicate lymphoid-biased hematopoiesis. We sincerely appreciate the valuable input, which helped us to further clarify the interpretation of our results.

Comment #3-8: Do the authors have any explanation for the high level of variabilitywithin the recipients of Hoxb5+ cells in Figure 2C?

Response #3-8:

We appreciate the reviewer's comment on this point. As noted in our previous report, transplantation of a sufficient number of HSCs results in stable donor chimerism, whereas a small number of HSCs leads to increased variability in donor chimerism1. Additionally, other studies have observed high variability when fewer than 10 HSCs are transplanted2-3. Based on this evidence, we consider that the transplantation of a small number of cells (10 cells) is the primary cause of the high level of variability observed.

Comment for our #3-8 response:

I agree that transplanting low number of HSC increases the mouse-to-mouse variability. For that reason, a larger cohort of recipients for this kind of experiment would be ideal.

Response:

Thank you for the insightful comment. We agree that a larger cohort of recipients would be ideal for this type of experiment. In Figure 2, the difference between Hoxb5+ and Hoxb5⁻ cells are robust, allowing for a clear statistical distinction despite the cohort size. However, we also recognize that a larger cohort would be necessary to detect more subtle differences, particularly in Figure 3. In response, we have added the following statement to the main text to acknowledge this limitation.

P9, L200 These findings unmistakably demonstrated that mixed/bulk-HSCs showed myeloid skewed hematopoiesis in PB with aging. In contrast, LT-HSCs maintained a consistent lineage output throughout life, although subtle differences between aged and young LT-HSCs may exist and cannot be entirely ruled out.

Comment #3-10: Is Figure 2G considering all primary recipients or only the ones that were used for secondary transplants? The second option would be a fairer comparison.

Response #3-10:

We appreciate the reviewer's comment on this point. We considered all primary recipients in Figure 2G to ensure a fair comparison, given the influence of various factors such as the radiosensitivity of individual recipient mice[1]. Comparing only the primary recipients used in the secondary transplantation would result in n = 3 (primary recipient) vs. n = 12 (secondary recipient). Including all primary recipients yields n = 11 vs. n = 12, providing a more balanced comparison. Therefore, we analyzed all primary recipient mice to ensure the reliability of our results.

Comment for our #3-10 response:

I respectfully disagree. Secondary recipients are derived from only 3 of the primary recipients. Therefore, the BM composition is determined by the composition of their donors. Including primary recipients that are not transplanted into secondary recipients for is not the fairest comparison for this analysis.

Thank you for your comment and for highlighting this important issue. We acknowledge the concern that including primary recipients that are not transplanted into secondary recipients is not the fairest comparison for this analysis. In response, we have reanalyzed the data using only the primary recipients whose bone marrow was actually transplanted into secondary recipients.

**Author response image 7. sa4fig7:** 

Importantly, the reanalysis confirmed that the kinetics of myeloid cell proportions in peripheral blood were consistent between primary and secondary transplant recipients. We sincerely appreciate your thoughtful feedback, which has helped us improve the clarity.

Comment #3-11: When discussing the transcriptional profile of young and aged HSCs, the authors claim that genes linked to myeloid differentiation remain unchanged in the LT-HSC fraction while there are significant changes in the STHSCs. However, 2 out of the 4 genes shown in Figure S4B show ratios higher than 1 in LT-HSCs.

Response #3-11:

Thank you for highlighting this important point. As the reviewer pointed out, when we analyze the expression of myeloid-related genes, some genes are elevated in aged LT-HSCs compared to young LT-HSCs. However, the GSEA analysis using myeloid-related gene sets, which include several hundred genes, shows no significant difference between young and aged LT-HSCs (see Figure S4C in this paper). Furthermore, functional experiments using the co-transplantation system show no difference in differentiation capacity between young and aged LT-HSCs (see Figure 3 in this paper). Based on these results, we conclude that LT-HSCs do not exhibit any change in differentiation capacity with aging.

Comment for our #3-11 response:

The authors used the data in Figure S4 to claim that "myeloid genes were tended to be enriched in aged bulk-HSCs but not in aged LT-HSCs compared to their respective controls" (this is the title of the figure; line # 1326). This is based on an increase in gene expression of CD150, vWF, Selp, Itgb3 in aged cells compared to young cells (Figure S4B). However, an increase in Selp and Itgb3 is also observed for LT-HSCs (lower magnitude, but still and increase).

Also, regarding the GSEA, the only term showing statistical significance in bulk HSCs is "Myeloid gene set", which does not reach significance in LT-HSCs, but present a trend for enrichment (q = 0.077). None of the terms in shown in this panel present statistical significance in ST-HSCs.

Thank you for your valuable point. As the reviewer noted, the current title may cause confusion. Therefore, we propose changing it to the following:

[P52, L1331] “Figure S4. Compared to their respective young controls, aged bulk-HSCs exhibit greater enrichment of myeloid gene expression than aged LT-HSCs”